# The molecular make up of smoldering myeloma highlights the evolutionary pathways leading to multiple myeloma

Eileen M. Boyle[1,2,3], Shayu Deshpande[1], Ruslana Tytarenko[1], Cody Ashby [1,4], Yan Wang[1], Michael A. Bauer [1,4], Sarah K. Johnson[1], Christopher P. Wardell [1,4], Sharmilan Thanendrarajan[1], Maurizio Zangari[1], Thierry Facon[5], Charles Dumontet [2], Bart Barlogie[6], Arnaldo Arbini[3], Even H. Rustad[3], Francesco Maura [7], Ola Landgren [7], Fenghuang Zhan[1], Frits van Rhee[1], Carolina Schinke [1], Faith E. Davies[3], Gareth J. Morgan [3,9✉] & Brian A. Walker [8,9✉]

Smoldering myeloma (SMM) is associated with a high-risk of progression to myeloma (MM). We report the results of a study of 82 patients with both targeted sequencing that included a capture of the immunoglobulin and *MYC* regions. By comparing these results to newly diagnosed myeloma (MM) we show fewer *NRAS* and *FAM46C* mutations together with fewer adverse translocations, del(1p), del(14q), del(16q), and del(17p) in SMM consistent with their role as drivers of the transition to MM. *KRAS* mutations are associated with a shorter time to progression (HR 3.5 (1.5–8.1), $p = 0.001$). In an analysis of change in clonal structure over time we studied 53 samples from nine patients at multiple time points. Branching evolutionary patterns, novel mutations, biallelic hits in crucial tumour suppressor genes, and segmental copy number changes are key mechanisms underlying the transition to MM, which can precede progression and be used to guide early intervention strategies.

[1] Myeloma Center, University of Arkansas for Medical Sciences, Little Rock, AR, USA. [2] INSERM 1052/CNRS 5286 Cancer Research Center of Lyon, Lyon, France. [3] Perlmutter Cancer Center, NYU Langone Health, New York, NY, USA. [4] Department of Biomedical Informatics, University of Arkansas for Medical Sciences, Little Rock, AR, USA. [5] Service des maladies du sang. Hôpital Claude Huriez, Lille University Hospital, Lille, France. [6] Division of Hematology, The Mount Sinai Hospital, New York, NY, USA. [7] Myeloma Service, Department of Medicine, Memorial Sloan Kettering Cancer Center, New York, NY, USA. [8] Division of Hematology Oncology, Indiana University, Indianapolis, IN, USA. [9] These authors contributed equally: Gareth J. Morgan, Brian A. Walker. ✉email: Gareth.morgan@nyulangone.org; bw75@iu.edu

Smoldering multiple myeloma (SMM) is an asymptomatic plasma cell disorder, distinguished from monoclonal gammopathy of undetermined significance (MGUS) by a higher rate of progression to symptomatic multiple myeloma (MM)[1–3]. The nature of SMM as a discreet disease entity has been called into question because of the triphasic shape of the Kaplan–Meier curves associated with its progression to MM[4]. These curves have been interpreted as being consistent with a low-risk group similar to MGUS, a high-risk group of "early myeloma" who progress quickly, and an intermediate risk group the nature of which is uncertain. The International Myeloma Working Group (IMWG) re-defined SMM[5] to take account of a number of risk factors including serum-free light chain (SFLC) ratio[6,7], bone marrow tumor burden[8], and bone lesions on magnetic resonance imaging (MRI) to identify a high-risk group[3,9,10]. A group with an 80% 2-year progression-free survival (PFS) was identified that was re-defined as MM[8]. Here, to enhance historical comparison, we use the term Early myeloma (EM), to identify patients that fail to meet the current criteria but would have been defined as SMM previously.

Importantly, newly defined SMM retains significant variation in the time to progression to MM, the molecular basis for which is uncertain. Efforts to recognize acquired genetic factors that can explain this variation have focused on MM related genetic factors including del(17p), t(4;14), MYC translocations[11], gain(1q)[12,13], and GEP risk scores[14]. Next-generation sequencing (NGS) allowed a better understanding of the molecular drivers of MM but until recently has not been widely applied to SMM[11,15].

NGS data from MM showed that there is clear spatio-temporal genetic variation consistent with an important role for evolutionary mechanisms. Further, current theories suggest that the majority of the genetic changes necessary to give rise to MM are present at the SMM stage and that there is either no change or limited changes in the sub-clonal architecture on the transition of SMM to MM[15,16]. However, currently, there are little NGS data available to inform our understandings of early disease stages when the so-called "trunk" of the MM "tree" is developing, and which gives rise to the observed variation in outcome[17]. To define the key genomic drivers for SMM, their impact on the sub-clonal structure, and on the outcome, we have analyzed a cross-sectional study of SMM cases and combined it with a study of multiple sequential samples derived from the same patients.

Here we identify fewer NRAS and FAM46C mutations together with fewer adverse translocations, del(1p), del(14q), del(16q), and del(17p) in SMM consistent with their role as drivers of the transition to MM. KRAS mutations are independently associated with a shorter time to progression. In an analysis of change in clonal structure over time, branching evolutionary patterns, novel mutations, biallelic hits in crucial tumor suppressor genes, and segmental copy-number changes are key mechanisms underlying the transition to MM.

## Results

### Identifying significant genomic differences between SMM and MM. 
Genetic abnormalities that are more frequent in MM have been suggested to be drivers of progression. Thus, in order to determine the spectrum of genomic differences characterizing SMM, we performed targeted sequencing to interrogate known drivers and immunoglobulin (Ig) translocations in a set of 82 previously untreated SMM patients and compared the results to a published data set of 223 MM patients analyzed using the same sequencing approach (Table 1[18]).

### Copy-number abnormalities and translocation frequency differs between SMM and MM. 
Translocations, copy-number abnormalities (CNA), and mutations are known to be drivers of MM. Overall, 35% ($n = 29/82$) of SMM samples had a canonical translocation, a frequency that is identical to that seen in MM, 37% ($n = 82/223$, non-Yates adjusted, $\chi^2 = 0.008$, $p = 0.82$). The most common translocation in the SMM data set was the t(11;14) seen in 23% of cases ($n = 19/82$) followed by the t(4;14) (4.9%, $n = 4/82$), t(6;14) (3.7%, $n = 3/82$), and t(14;16) (2.4%, $n = 2/82$). Two samples were identified with a MAFB translocation [one t(14;20) and one t(20;22)]. There were significantly fewer t(4;14) among the SMM patients (non-Yates adjusted $\chi^2 = 4.4$, $p = 0.03$) and more t(11;14) (non-Yates adjusted $\chi^2 = 4.6$, $p = 0.03$), suggesting that SMM carried fewer adverse risk cytogenetic groups [t(4;14), t(14;16), t(14;20)] than MM (9.8% versus 20%, non-Yates adjusted $\chi^2 = 4.5$, $p = 0.03$) (Fig. 1a).

The genomic partners of translocations and secondary rearrangements detected at the MYC locus were similar in the two datasets but differed in frequency. Rearrangements involving MYC were seen in 35% ($n = 29/82$) of SMM cases, which is fewer than were found in MM (55%, $n = 124/223$, $\chi^2 = 6.9$, $p = 0.009$). Fifty-five percent of the rearrangements ($n = 16/29$) involved a translocation with the remainder being either duplications, inversions, or gain of a region surrounding 8q24. Most translocations to MYC involved non-Ig partners (11/16, 69%), with the remainder involving IGH (3/5) or IGL (2/5). This frequency is similar to that seen in MM (63% non-Ig).

In SMM, among the other translocations (non-canonical Ig translocation, and non-Ig-MYC translocations), the two recurrent partners identified were FAM46C ($n = 2$) and TXNDC5 ($n = 2$), which are also the most common recurrent partners in MM ($n = 6$ and $n = 4$, respectively). Some translocation partners were specific to SMM or MM, but they were non-recurrent and not reflective of a significant difference (Fig. 1b).

CNA have been shown to be prognostically important in MM and to accurately determine their spectrum in SMM, we performed ultra low-pass whole-genome sequencing (ULP-WGS) on a subset of patients (69 patients with SMM and 116 newly diagnosed MM (NDMM) patients) to determine CNA across the genome. The recurrent regions identified were similar to those seen in MM (Fig. 1c). Using the targeted panel on the whole data set, CNAs which were less frequent in SMM included del(1p) (FAF1/CDKN2C; 2% vs. 17%, $\chi^2 = 11.2$, $p = 0.0008$), del(8p) (NSD3) (3.6% vs. 15%, $\chi^2 = 6.5$, $p = 0.01$), del(14q) (TRAF3) (7% vs. 19%, $\chi^2 = 5.5$, $p = 0.02$), del(16q) (CYLD) (13% vs. 28%, $\chi^2 = 5.8$, $p = 0.01$), and del(17p) (TP53) (6% vs. 15%, $\chi^2 = 4.0$, $p = 0.04$) (Supplemental Table 1).

### The frequency of known driver gene mutations is greater in MM. 
We next investigated the frequency of important mutations in SMM and MM using the same targeted sequencing panel. We applied analysis of a catalog of key MM mutations[18] to cases with MGUS, SMM, EM, and MM. There was a significant difference in the overall number of mutations identified at various disease stages (Supplemental Fig. 1).

Overall, 61% and 82% of SMM and MM had a mutation in one of the previously described mutational drivers that were present on the panel (Supplemental Table 3[19]). There were no IDH2, MAX, XBP1, CDKN2C, RB1, and SET2D mutations in the SMM group. The genomic abnormalities identified in each SMM sample are summarized in Fig. 2. The most frequently mutated gene in SMM was KRAS ($n = 11$, 13.4%). In comparison to NDMM, fewer NRAS (4.5% versus 17%, $\chi^2 = 6.4$, $p = 0.01$), and FAM46C (0% versus 7%, Fisher's, $p = 0.008$) mutations were detected and there was a trend towards fewer KRAS (13% vs 22%) mutations in SMM (Fig. 2a and Supplemental Fig. 2). A comparison of rare mutations was not possible given the sample

**Table 1 Summary of patient's characteristics.**

| Characteristic | SMM (n = 82) | MM (n = 223) |
|---|---|---|
| Median age at diagnosis (range) | 63 years (37–85) | 59 years (30–75) |
| Median age at sample collection (range) | 65 years (38–85) | 59 years (30–75) |
| Median follow-up (95% CI) | 5.18 years (95% CI 3.53–6.89) | 8.14 years (95% CI 7.35–8.96). |
| 5-year progression-free survival (95% CI) to symptomatic MM | 69.9% (95% CI 58–83%) | NA |
| 5-year overall survival (95% CI) | 96.5% (95% CI 92–100%) | 6.17 years (95% CI 5.18–7.75) |
| Timepoint | Diagnosis: 44% | Diagnosis: 100% |
| | Follow-up: 56% after a median time 2.9 years from diagnosis | |
| Gender ratio (female:male) | 1:1.15 | 1:1.8 |
| Race | 24% African-American | 10% African-American |
| | 71% White-Caucasian | 88% White-Caucasian |
| | 5% Other (Native American, Pacific Islander or unknown) | 2% Other (Native American, Pacific Islander or unknown) |
| Isotype | IgA: 30% (n = 25) | |
| | IgG: 60% (n = 49) | |
| | IGD: 1.2% (n = 1) | |
| | LCO: 8% (n = 7) | |

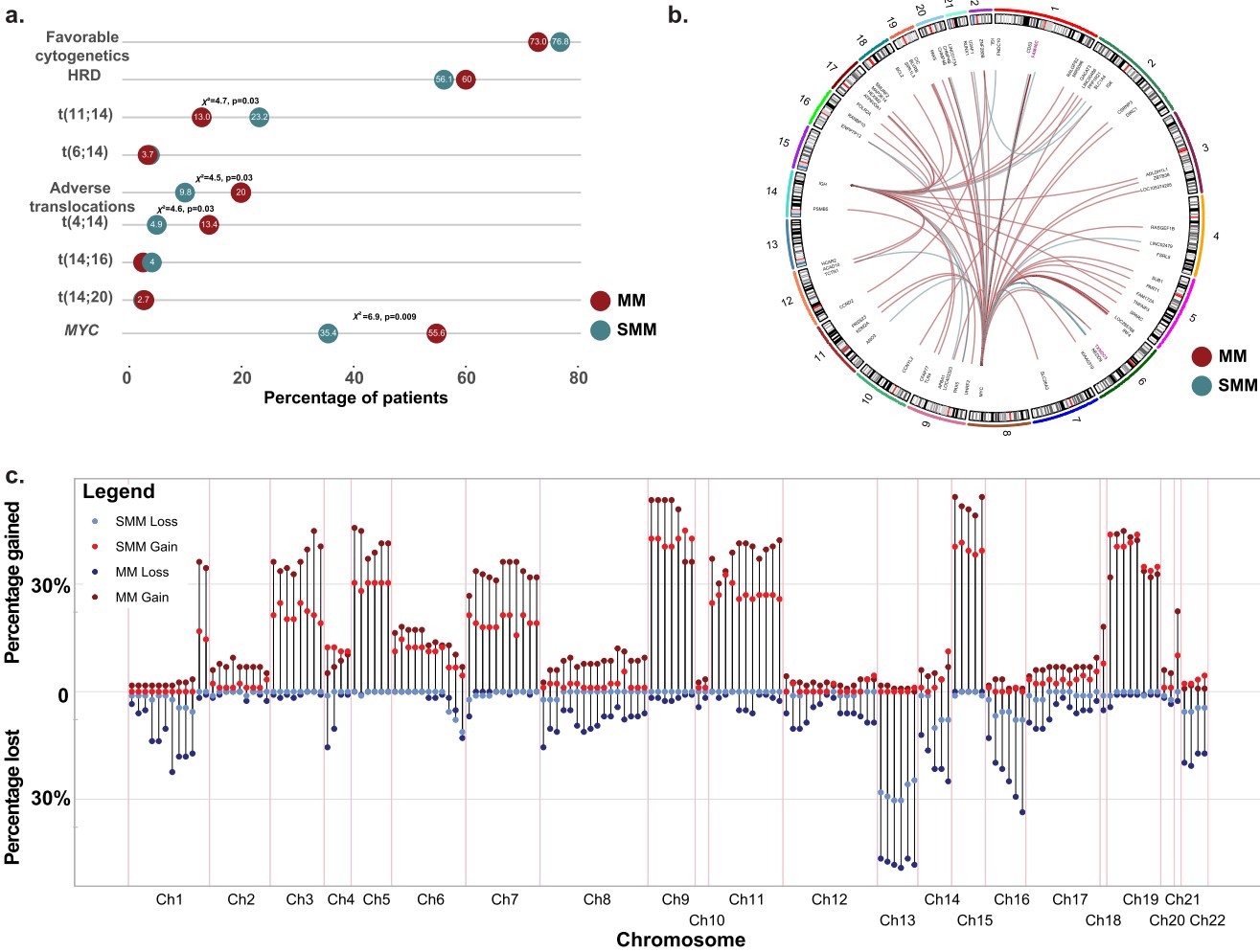

**Fig. 1 Structural events in smoldering multiple myeloma (SMM) compared to newly diagnosed myeloma (MM). a** Frequency of cytogenetic subgroups and main translocations in SMM and MM suggesting that fewer SMM have high-risk features. **b** Circos plot of the landscape of non-Ig-*MYC* and non-canonical Ig translocations highlighting *FAM46C* and *TXNDC5* as current recurrent partners. **c** Copy-number changes in 160 recurrently altered regions covering the known drivers (Supplemental Table 4) in SMM and MM suggesting the landscape is similar but there are fewer copy-number abnormalities. Adverse translocations = t(4;14), t(14;16), t(14;20). Favorable cytogenetics = t(11;14), HRD.

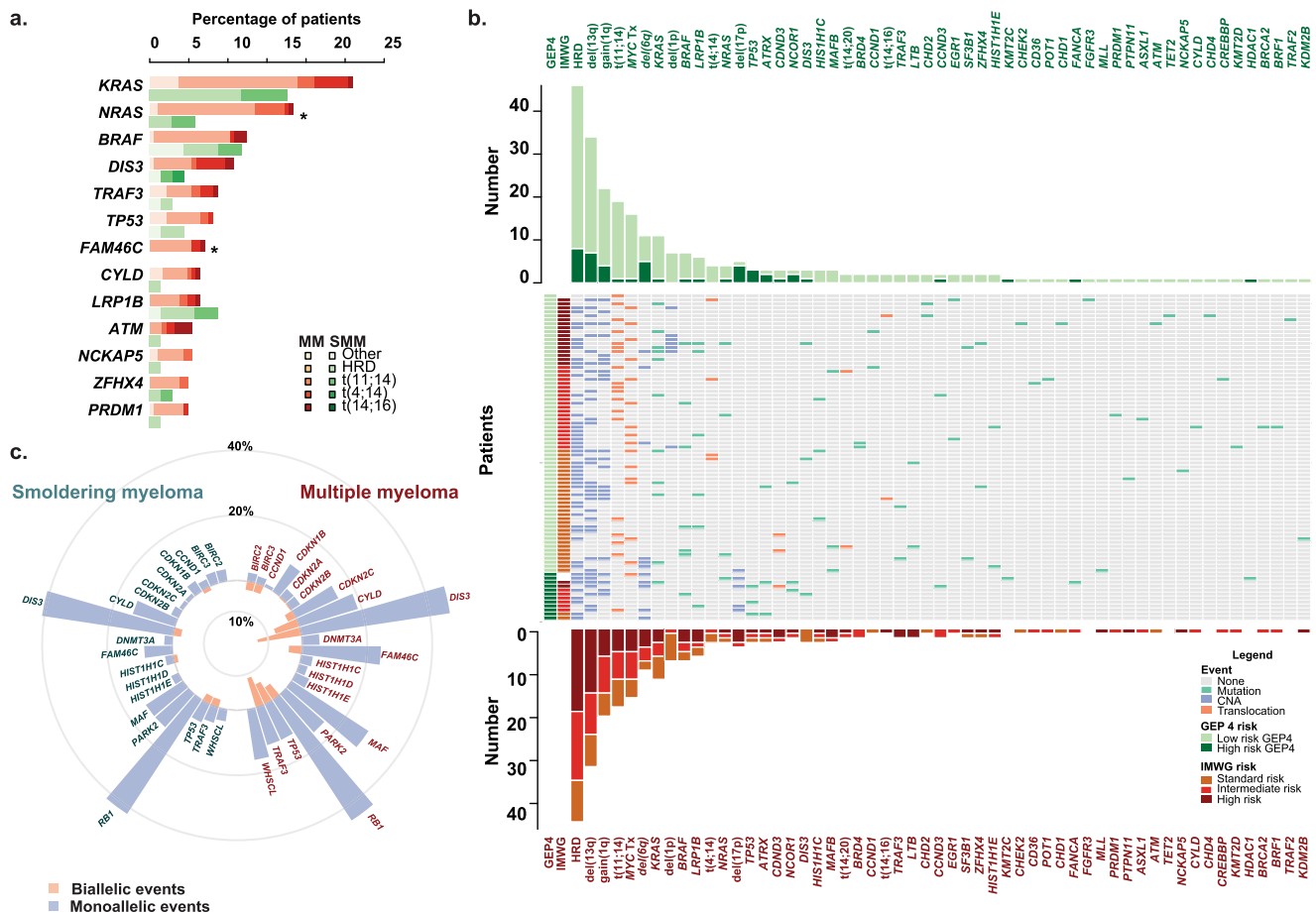

**Fig. 2 Mutational events in smoldering myeloma in comparison to myeloma. a** The most frequently mutated genes and their distribution across the most common molecular subgroups. **b** The distribution of somatic abnormalities per sample and risk group. **c** Frequency of biallelic and monoallelic events per driver locus in SMM and MM showing fewer biallelic drivers in SMM. *Significantly different at $p < 0.05$ level, two-sided Fisher's test.

size and so we restricted our analysis to the comparison of mutations occurring in known myeloma relevant pathways. Overall, we identified a significant difference in the frequency of mutations in the MAPK pathway (*BRAF*, *NRAS*, or *KRAS* mutation) between SMM and MM (24% vs. 44%, $\chi^2 = 9.2$, $p = 0.002$). There were also fewer patients with NF-κB pathway mutations (*BIRC2*, *BIRC3*, *BCL10*, *CYLD*, *IRF4*, *MAP3K14*, *TRAF2*, or *TRAF3*; 5% vs. 16%, $\chi^2 = 5.7$, $p = 0.02$) and a trend suggesting fewer DNA repair pathway mutated patients (*TP53*, *ATM*, *ATR*, *ATRX*, *BRCA1*, or *BRCA2*; 7% vs. 17%, $\chi^2 = 3.8$, $p = 0.0504$).

In terms of mutational signatures, we show that the contribution of APOBEC is significantly lower in SMM than in MM (11% (0–43%) versus 17% (0–48%), $\chi^2 = 5.2$, $p = 0.02$) consistent with it having a role later in disease progression, Fig. 3a. Furthermore, in SMM patients with either a *MAF* or *MAFB* translocation (termed maf), the median APOBEC contribution is 18% (0–54%, $n = 4$), compared to 11% (0–43%, $\chi^2 = 0.5$, $p = 0.4$, $n = 78$) in non-maf samples. Therefore, unlike observations in MM, we do not identify any significant difference between the two subgroups in SMM. Finally, the APOBEC contribution also seems lower in maf-SMM than it does in maf-MM (16% (0–44%, $n = 4$) vs. 41% (0–100%, $n = 15$)). Despite the small sample size, these data suggest that APOBEC is associated with disease progression (Fig. 3a and Supplemental Fig. 3A).

**Biallelic inactivation of tumor suppressor genes is less frequent in SMM.** Tumor suppressor gene inactivation is an important

mechanism of relapse in NDMM with biallelic events having the most penetrant effects[20]. To determine the role of this mechanism in SMM, we defined a list of 20 previously identified tumor suppressor gene loci of relevance to MM at relapse and investigated them for biallelic inactivation using combined mutation and CNA analysis. In total there were 103 biallelic events identified in MM (103 in 64/223 patients) versus only eight in SMM (8 in 8/82 patients) ($\chi^2 = 10.9$, $p = 0.001$) suggesting second hits at the same locus are a hallmark mechanism of the transition to MM (Fig. 2c). One double-hit patient, defined by a biallelic *TP53* inactivation was identified in this series of SMM patients compared to 18 in MM. Other key biallelic events included *DIS3, RB1, FAM46C,* and *TRAF3*. Interestingly, biallelic *DIS3*, which has been associated with an adverse outcome in MM[18], was present in 5% of MM cases vs. only 2% of SMM cases, again consistent with it being associated with a more aggressive disease phenotype. These results are consistent with data previously generated at relapse after treatment and indicate that biallelic inactivation of tumor suppressor genes is an important mechanism of progression at all disease stages.

**Mutations in *KRAS* are associated with a shorter time to progression.** In order to determine the molecular factors associated with time to progression to MM, we assessed outcome in 77 patients evaluable for progression. Thirteen percent of patients were identified as high-risk according to the GEP4 risk-score[15], 23% were high-risk and 35% intermediate risk according to IMWG criteria. With a median follow-up of 5.18 years, we

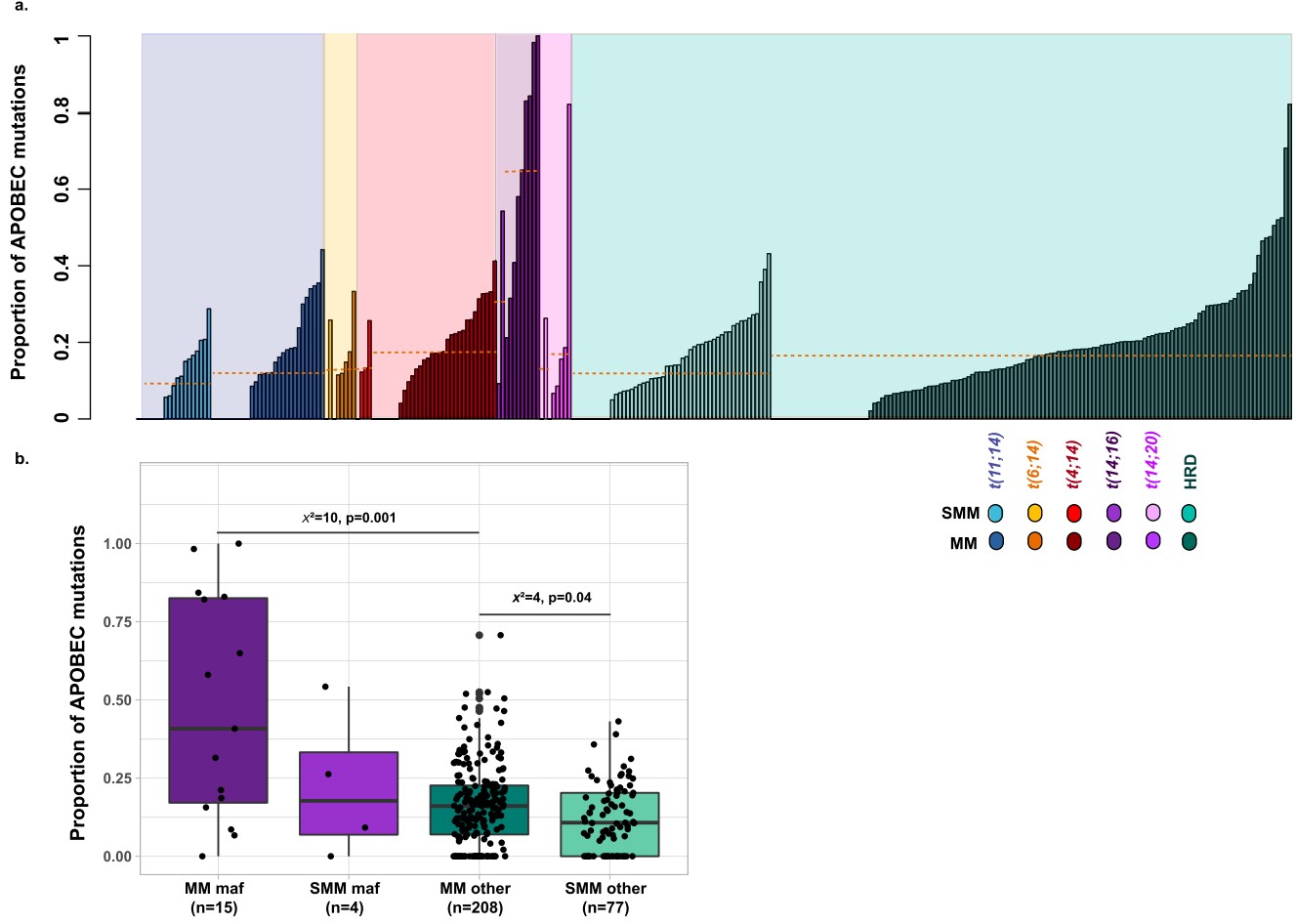

**Fig. 3 Contribution of APOBEC signatures to the mutational landscape of SMM and MM. a** Contribution of the APOBEC signature in SMM and MM by cytogenetic subgroup (yellow lines = median). **b** Contribution of the APOBEC signature in SMM and MM in maf and non-maf samples. Boxplots representing second quartile, median, third quartile, whiskers representing first, and fourth quartile. All data points including outliers are represented. $X^2$ = chi-squared, two-sided $p$-value derived from Kruskal–Wallis test, $n$ = number of patients.

identified a 5-year progression rate of 30.1%, with no plateau consistent with an ongoing risk of progression (Fig. 4a).

To determine the associations with high-risk SMM, we analyzed the distribution of mutations across the IMWG and GEP4 risk groups (Fig. 2b and Supplemental Fig. 4). Although the numbers were low, we note that *TP53* ($n = 3/3$) mutations and del(17p) ($n = 3/4$) were enriched in the high-risk GEP4 patients. Mutations in *DIS3* ($n = 3$), *CCND1* ($n = 2$), and *ATM* ($n = 1$), as well as the t(4;14) were associated with the IMWG definition of high-risk.

In a univariate analysis of time to progression to MM including all abnormalities present in at least seven samples (Fig. 4b), we show that high-risk IMWG status and the GEP4 risk were associated with an adverse outcome (Supplemental Fig. 5B, C). Importantly, the presence of a *KRAS* mutation ($n = 11$) was associated with a short time to progression (HR 3.5 (1.5–8.1) $p = 0.002$) (Fig. 4c). Del(6q) ($n = 9$) was also associated with a short time to progression (HR 4.5 (1.9–11), $p = 0.0005$) (Supplemental Fig. 5A). Of note, four of these patients had high-risk status according to GEP4, and two of them carried a del(17p). Biallelic events of driver genes were not associated with outcome (data not shown). A combined analysis of the impact of *BRAF*, *KRAS*, and *NRAS* on time to progression was not associated with a significant impact (Supplemental Fig. 5D). The presence of a *MYC* translocation has previously been identified as an adverse prognostic factor in SMM[11,21]. Here, in a set of patients defined

by the current IMWG criteria, we did not find that *MYC* translocations were associated with adverse outcome[8]. Finally, there was a trend suggesting that patients with a small contribution of APOBEC signatures (<5%) had a better outcome than the others (Supplemental Fig. 3B).

**Multivariate analysis of molecular markers involved in time to progression.** In order to identify independent prognostic markers a multivariate analysis using previously published risk scores (GEP4 and IMWG) alongside factors occurring at a frequency of $n \geq 10$ and at least a borderline impact on univariate time to progression (*KRAS* mutations and del(13q)) was performed. The presence of a *KRAS* mutation, a high-risk GEP4, or high-risk IMWG retained their impact on outcome (Fig. 4d). Indeed, the presence of a *KRAS* mutation, a high-risk GEP4, and high-risk IMWG was associated with a hazard ratio for progression of 3.2 (CI 1.3–8.1, $p = 0.011$), 5.3 (CI 1.6–18.2, $p = 0.008$), and 3.1 (CI 1.4–7.2, $p = 0.007$), respectively, consistent with their independent contribution to risk at this disease stage. Thus, we show in a significant number of cases the important prognostic contribution of mutations in *KRAS* at the SMM disease stage. This observation contrasts significantly with the lack of association with a prognosis that RAS mutations have in NDMM and highlights their important role in the transition from asymptomatic to symptomatic stages of the disease. We also performed

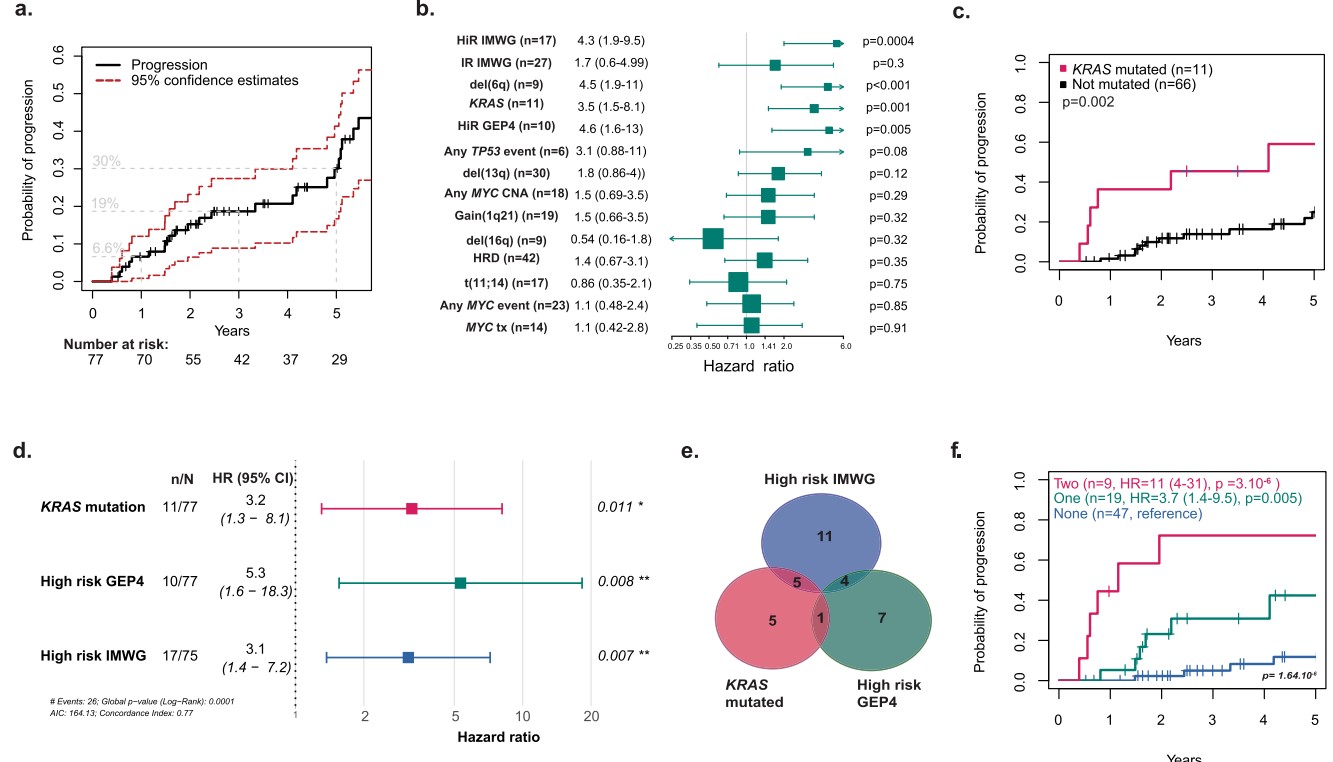

**Fig. 4 Prognostic impact of molecular and clinical features in SMM. a** Progression-free survival with a 30% progression rate at 5 years with no plateau, suggesting an ongoing risk. **b** Univariate analysis of molecular features in SMM. **c** Impact of *KRAS* mutations on the outcome. **d** Multivariate analysis suggests GEP4, high-risk IMWG, and *KRAS* mutations are independent prognostic factors. **e** Venn diagram showing the overlap between high-risk GEP4, high-risk IMWG, and *KRAS* mutations. **f** Progression-free survival for patients with either none, one, or two adverse factors defined as high-risk GEP4, high-risk IMWG, and *KRAS* mutations suggesting these factors are additive. IMWG International Myeloma Working Group, HiR high-risk, IR intermediate risk, HRD hyperdiploid, CNA copy-number abnormality, tx translocation, GEP4 4 gene expression profile, *n* number of patients with event, *N* total number of patients evaluable, error bars = 95% CI.

the analysis with variants present at a number of $n \geq 7$ and found similar results (Supplemental Fig. 5E).

To determine whether *KRAS* mutations have the potential to contribute to novel risk scores, we compared the *KRAS* mutated cases with the two previously used risk models and show that only 13% of patients overlapped between one of the three risk models, with five patients having only a *KRAS* mutation (Fig. 4e). No patient fell into the intersection of all three risk factors suggesting that enumerating these in clinical cases may contribute useful additional prognostic information.

We have shown previously that an accumulation of risk factors leads to higher risk status in MM[22], and the results of a similar analysis confirm that the same is true in SMM. When more than one molecular lesion was present, an HR of 11.2 (4–31) for progression was seen with an associated median time to progression of 1.16 years (Fig. 4f), compared to an HR of 3.7 (1.4–9.5) and 5-year PFS of 57% (35–94%) when only one risk factor is present and an 88% (78–99%) 5-year PFS when none are present. This was clearer when considering the patients whose samples were taken within the first 90 days of diagnosis (Supplemental Fig. 6).

**Pathway deregulation associated with progression.** Mutations of the NF-κB pathway are among the most frequent in plasma cell disorders making an understanding of their impact in SMM critical. Mutations affecting NF-κB were seen in 5% of SMM patients, which is lower than the 16% seen in MM patients ($\chi^2 = 5.7$, $p = 0.02$). Mutations in the NF-κB pathway did not associate with outcome in this data set (data not shown). Based on

previously published work[23], we generated gene signatures, from gene expression array data, that correlated with NF-κB p65 activity that can be used as a surrogate for changes in NF-κB pathway activity. These results show that the score is not statistically different in SMM compared to MM [SMM-10.7 (IQR: 9.6–11.9) versus MM-10.57 (IQR (8.68–12.6) $p = 0.8$)]. However, it was lower in normal CD138$^+$ cells (11.4 (11–11.7), $p < 0.0001$). These data suggest that NF-κB dependency is similar in SMM and MM, consistent with dysregulations in this pathway being early events associated with the development of MGUS and not with the transition of SMM to MM (Supplementary Fig. 7).

**Sequential molecular changes identified within individual cases over time.** In order to investigate the temporal relationship of genetic changes associated with progression, we studied 53 sequential samples obtained from nine patients with a median follow-up of 7.26 years (5.17–∞). Six of the patients progressed during the time of follow up, with progression samples being available for 5/6 patients. Two patients with SMM had not progressed at the time of the analysis. Additionally, one of the 9 patients, patient G, who had features of early myeloma, progressed to MM within the first 6 months (Fig. 5a). We performed WES to a depth of 93× to study changes in the molecular architecture of the clone over time.

**Structural abnormalities.** We identified four patients with a t(11;14) (patients C, D, E, F) and one with a t(4;14) (patient I). These molecular events were detected as being clonal in each of the sequential samples. Five patients had hyperdiploidy (HRD)

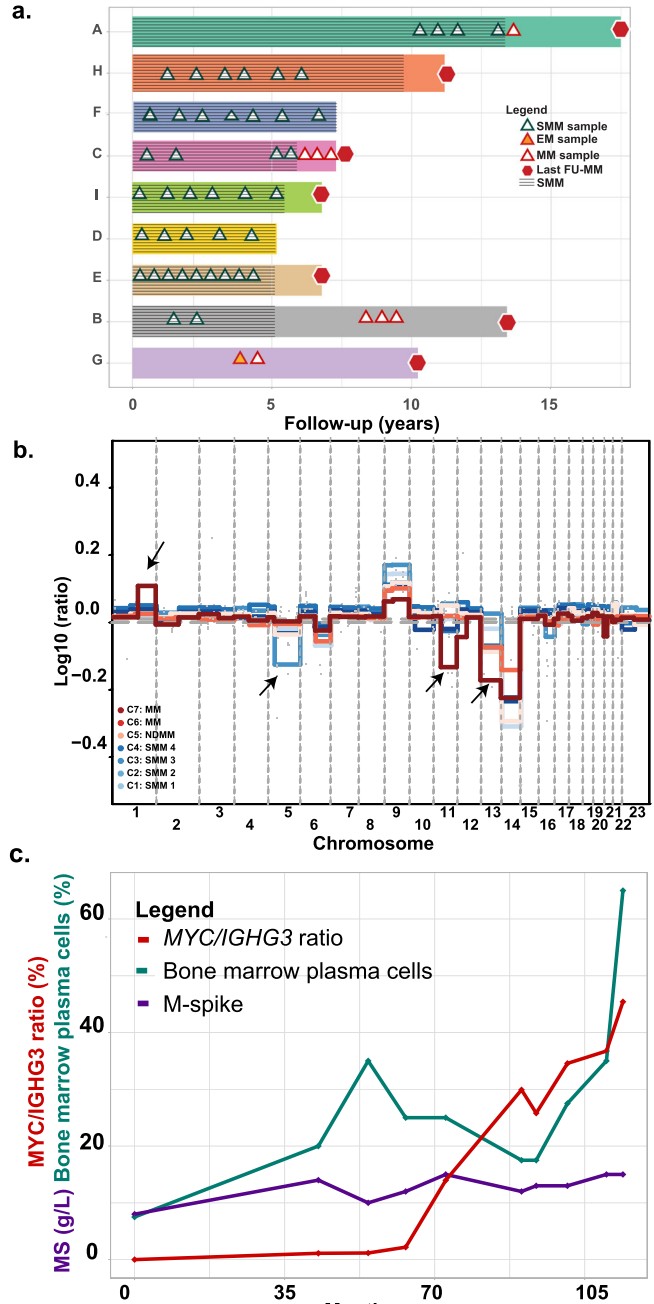

**Fig. 5 Acquisition of drivers in SMM patients over time. a** Swimmer plot of the group of patients (A–G) followed over time. The color bars represent progression-free survival. **b** Plot showing changes in copy number over time with a focus on the loss of del(5) and the acquisition of gain(1q), del(11q), and del(13). Arrows highlight changes in CNA. **c** The acquisition of a t(8;14) within a myeloma propagating cell leads to outgrowth of the clone until it dominates the tumor population. EM early myeloma, FU follow-up, NDMM newly diagnosed myeloma, NGS next-generation sequencing, MS M-spike.

While only limited numbers of segmental CN change were identified, some provide informative examples. One patient for example acquired a *DIS3* and *RB1* deletion through the loss of chromosome 13, which gradually become more clonal over time, as well as a gain(1q) and a del(11q) at progression (Fig. 5b). The sequential data provided additional insights into the role played by CN change allowing the assessment of changes over time. In this fashion, we were able to identify fluctuations in the levels of individual clones defined by CN at specific sites that are not seen at the transition to MM. These include a loss of chromosome 5 in patient C that was not sustained at progression (Fig. 5b) and a loss of 16q that was only seen in patient B's initial sample (Supplemental Fig. 8).

Importantly, we identified a case where a t(8;14) appeared during follow-up allowing us to determine in an individual case its impact on sub-clonal size. The translocation involved a rearrangement between *MYC* and the *IGHA2* switch region developing at a relatively early stage of disease; post-initial immortalization but before clinical myeloma. Using ddPCR, we quantified the rearranged Ig-*MYC* allele and the productive *IGHG3* allele. This analysis confirmed that the translocation was not present at diagnosis, appeared in a small fraction (1%) 3.0 years after initial diagnosis, and steadily increased over time, reaching 45% of cells on the last sample, 8.2 years from the initial diagnosis. This increase was matched by an increase in bone marrow plasma cells and M-spike (Fig. 5c). These findings are consistent with it having contributed to clonal expansion potentially via an increased proliferation rate.

**Mutational load in SMM patients increases over time**. The mutational load has the potential to both contribute to progression and to be a marker of changes in mutational mechanisms. To investigate changes in mutational load in this set of serial samples and to account for variation in depth of coverage, we normalized the number of mutations per sample according to the depth of coverage and analyzed trends in mutational load over time. At the SMM stage, the mutational load increased with time (Fig. 6a). Samples with hyperdiploidy (HRD) seemed to have a higher mutational rate than other subgroups (nHRD) but the follow-up was longer in this group of patients (Supplemental Fig. 9A). The mutational rate of patients that eventually progressed was not statistically different from the mutation rate of those that have not progressed (Supplemental Fig. 9B), but there was a trend suggesting the mutation rate increased around progression in SMM (Fig. 6b).

Focussing on previously described[19] mutational drivers, we identified a median of 1.5 (range 0–4) drivers per SMM sample with 21 genes being mutated in 31/53 of samples. We identified a correlation between the number of drivers present and time from the clinical presentation but this was driven by one sample only (A) (Fig. 6c). The Cancer Clonal fraction (CCF) of the driver mutations increased over time (Fig. 6d), consistent with them being actively selected. These findings are consistent with prior data derived from paired SMM/MM studies where only limited numbers of new mutational drivers at progression were seen.

**Mutational processes are stable at the SMM/MM interface**. Signatures of mutational processes have emerged as an important tool to determine both intrinsic and extrinsic mechanistic factors mediating cancer etiology and progression. Here, we examined the role played by mutational signatures at the SMM disease phase in serial samples. The background signatures (SBS1 and SBS5)[24] are the major contributors to the mutational patterns seen in SMM. A single patient had a non-canonical AID signature (SBS9) seen in more than one sample, on an HRD background

(patients B, E, F, G, I). Neither an etiological translocation nor hyperdiploidy could be detected in patient A but there was evidence of segmental copy-number (CN) changes. We have previously defined a catalog of segmental CNA in NDMM that contribute significantly to prognosis. In this series of sequential samples, we show that hyperdiploidy is an early event and is stable over time and does not constitute a significant mediator of progression. In contrast, we identified a number of regions of segmental CN change that may contribute to disease progression.

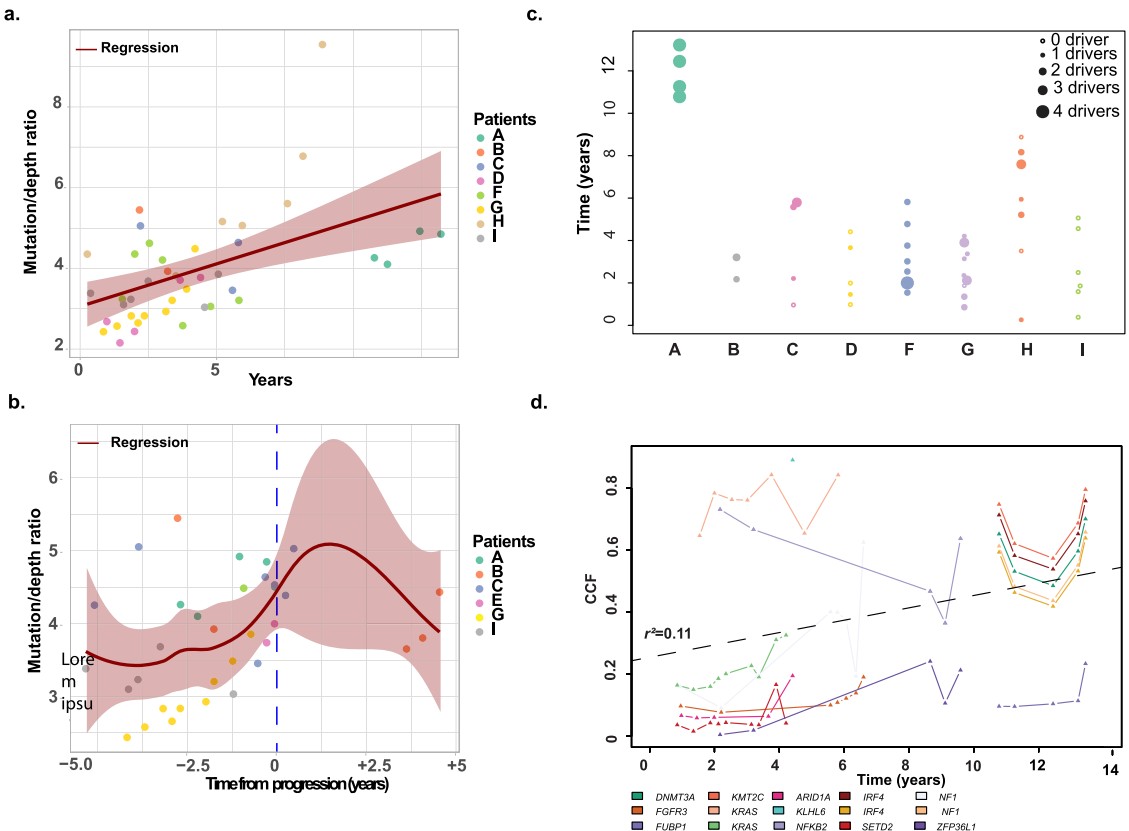

**Fig. 6 Mutational changes over time. a** The mutation rate of patients increased over time. **b** Progression rate among progressors suggesting that the mutation rate is high at the SMM stage around progression. **c** Evolution of the number of mutational drivers per sample over time suggesting fluctuation but no steady increase. **d** Evolution of the CCF of each driver mutations over time. Error bands = 95% CI.

(data not shown). To determine whether changes in mutational processes correspond with the transition to NDMM, we analyzed the contribution of each signature in relation to the time of progression to MM. Samples taken more than 2 years prior to progression had the same signatures as both those taken within 2 years of progression and those cases that did not progress (Fig. 7a). This observation supports the idea that mutational processes are stable over time and do not define the transition to MM.

In order to further dissect out the contribution of mutational signatures, we clustered mutations into five groups based on their respective CCF. Overall, the contribution of mutational signatures did not vary substantially according to clonality (Fig. 7b). We identified mutations in genes with a CCF that increased from intermediate (CCF 0.4–0.8) to clonal (0.8–1) such as *KRAS, CHD2, ABCC2, TNIP1, TRPS1, RCCD1, MTRR, ERCC6*, and from low CCF (0–0.4) to medium (0.4–0.8) such as *RMI1, PSMA8 CAMK1D, GALNT2, PON3, SALL4*, that could be potential drivers (Fig. 7c).

**Changes in sub-clonal architecture precede progression.** The clinical management of SMM is different from NDMM in that it is possible to monitor progression over time and to reassess the disease on a number of occasions without impairing clinical outcomes. Therefore, we determined whether it may be clinically relevant to monitor sub-clonal structure as a means of predicting progression. We reconstructed sub-clonal structure and followed the size of each clone over time using Pyclone in eight patients. A median of eight clones per patient was identified, most related via branching patterns (7/8) with only one case being associated with

a linear pattern. Five clones made up 90% of the tumor in 84% of cases, and six clones in 16% of cases. In the year prior to biochemical progression, significant changes in sub-clonal structures were seen in all evaluable patients (Fig. 8 and Supplemental Figs. 10–14). One patient that progressed was not evaluable as the pre-progression sample was 5 years earlier (Supplemental Fig. 15). These data suggest that monitoring sub-clonal structure as a means of disease assessment is possible and could direct clinical intervention at a time long before end-organ damage or clinical symptoms develop, highlighting the importance of bone marrow monitoring of SMM patients.

**Clonal diversity is a marker of time to progression.** These early disease stages can be considered as forming a distinct ecosystem and as such can be studied using ecological tools. The Shannon Diversity index[25] is one such tool that has been extensively used and we applied this approach to each of the sequential SMM samples. This analysis showed that samples from patients that progressed had a higher index than those that did not ($\chi^2 = 11$, $p = 0.0006$). There was no difference between patients with *KRAS* mutations or high-risk features (t(4;14) and gain(1q)). However, we went on to show that there was a trend for patients that progressed to have a stable index while those that did not progress had evidence of ongoing change. These observations are consistent with a high diversity index that is stable being associated with already transformed disease (Supplemental Fig. 16). In contrast, in cases of SMM, an increase in diversity was seen over time. These observations suggest that applying the *H*-indices to serial samples may be able to identify cases at higher risk of progression but need confirmation in larger datasets.

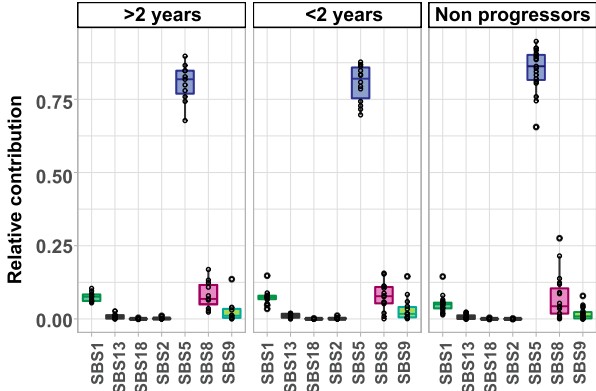

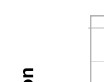

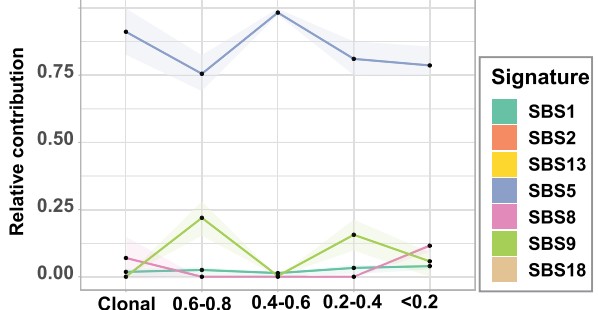

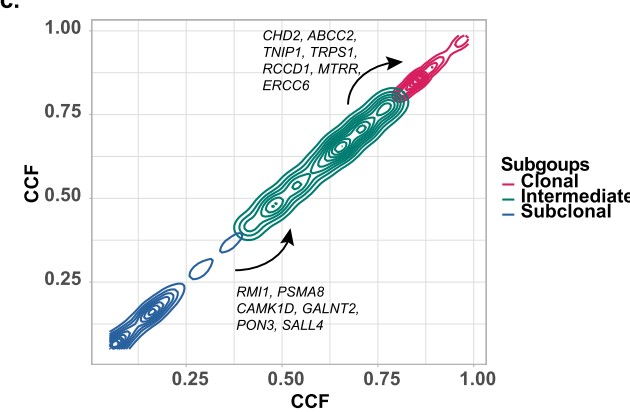

**Fig. 7 Mutational signatures.** Mutational signatures. **a** The mutational signature composition does not differ between patients that progress more or less than 2 years after the sample was taken and those who do not progress. Boxplots representing second quartile, median, third quartile, whiskers representing first, and fourth quartile. All data points including outliers are represented. **b** The contribution of mutational signatures did not vary substantially according to clonality. **c** The CCF of genes increased overtime suggesting they may be drivers. HRD hyperdiploid, nHRD non-hyperdiploid, SBS single base substitution, CCF cancer clonal fraction. Error bands = 95% CI.

## Discussion

Managing SMM is a major clinical goal and the key to doing this effectively is to define the optimum time for an intervention. Using genetic variants to identify cases of SMM at substantial risk of transformation before the development of end-organ damage is a key aim. Further, adding genetic factors into personalized risk-assessment tools may improve treatment initiation decisions.

Utilizing a priori information on drivers of MM and analyzing their impact in SMM, we suggest that a subset of these molecular events, including *KRAS*, *NRAS*, and *FAM46C*[26], are MM defining events, which when present identify cases in the process of transformation. This conclusion is supported by the analysis of the interaction of the variants with a sub-clonal structure based on a unique series of sequential samples from SMM where there is ongoing sub-clonal expansion ultimately leading to a change in clinical behavior. An alternative interpretation may be that high-risk MM evolves more rapidly through a shorter smoldering phase thus explaining why it is under-represented in SMM. In contrast to the genetic hits outlined above, NF-κB dependency at this disease stage does not have an impact on PFS or sub-clonal expansion. Thus, while these events occur at early molecular timepoints and contribute to the genetic complexity of MGUS, they are not associated with symptomatic behavior.

Further, from a clinical perspective, we identify an important role for *KRAS* mutations as a prognostic factor that can significantly contribute to risk assignment in SMM[27]. With a hazard ratio for progression of 3.8, *KRAS* mutations are a critically important molecular prognostic factor. A recent study[21], identified a similar impact confirming the prognostic relevance of these molecular variants. It is important to consider why such a prognostic effect is seen in SMM but not in MM and this may reflect the impact on cell-cycle progression promoting a clonal sweep and transition to MM. Thus, cases with *KRAS* mutations may be better defined as MM in the process of transition that are better grouped with MM rather than MGUS. Consistent with this hypothesis, prior studies using the plasma cell labeling index have found significant differences between precursor phases and NDMM[28].

Further evidence for the role of a proliferation advantage in symptomatic behavior and with time to progression in SMM comes from our identification of a case of SMM, which acquired a translocation at the *MYC* locus. The CCF of the subclone, defined by this abnormality, sequentially increased over time. We were, however, unable to identify a prognostic impact of *MYC* rearrangements, as has been suggested previously[11,21]. We speculate that this difference is a reflection of the exclusion of many such events in this study by the application of the stringent new diagnostic definitions of SMM. Consistent with this idea, the current study had a lower incidence of *MYC* translocated cases than was seen in other previous series[29].

Knowledge of the role of structural events in the precursor stages of the disease has been limited. Here, we show that one of the major initiating events of MM, HRD, is stable and does not constitute a significant mechanism impacting time to progression. In contrast, segmental CN gains and losses together are seen to fluctuate over time consistent "clonal tiding" of sub-clones defined by these abnormalities. In this data set, the sites of these events seem to be restricted to the sites of known tumor suppressor genes identified in MM such as on chromosome 16q, which is one of the most common features in MM at 30% but only in 18% of SMM. We identified significant differences in the frequency of CN changes between SMM and MM providing further support with their role as drivers. However, these differences may be explained by the different composition of molecular subgroups in the asymptomatic disease stages. For example, there are fewer t(4;14) patients in the SMM data set compared to MM, and therefore the frequency of CN changes associated with this translocation, such as del(16q), may be impacted by this.

Knowledge of the molecular events underlying the genetic complexity of the early truncal stages of MM has largely been extrapolated from the study of NDMM. Here, we have used sequential samples from the early disease stage of SMM to improve the resolution of the definitions of sub-clonal structure.

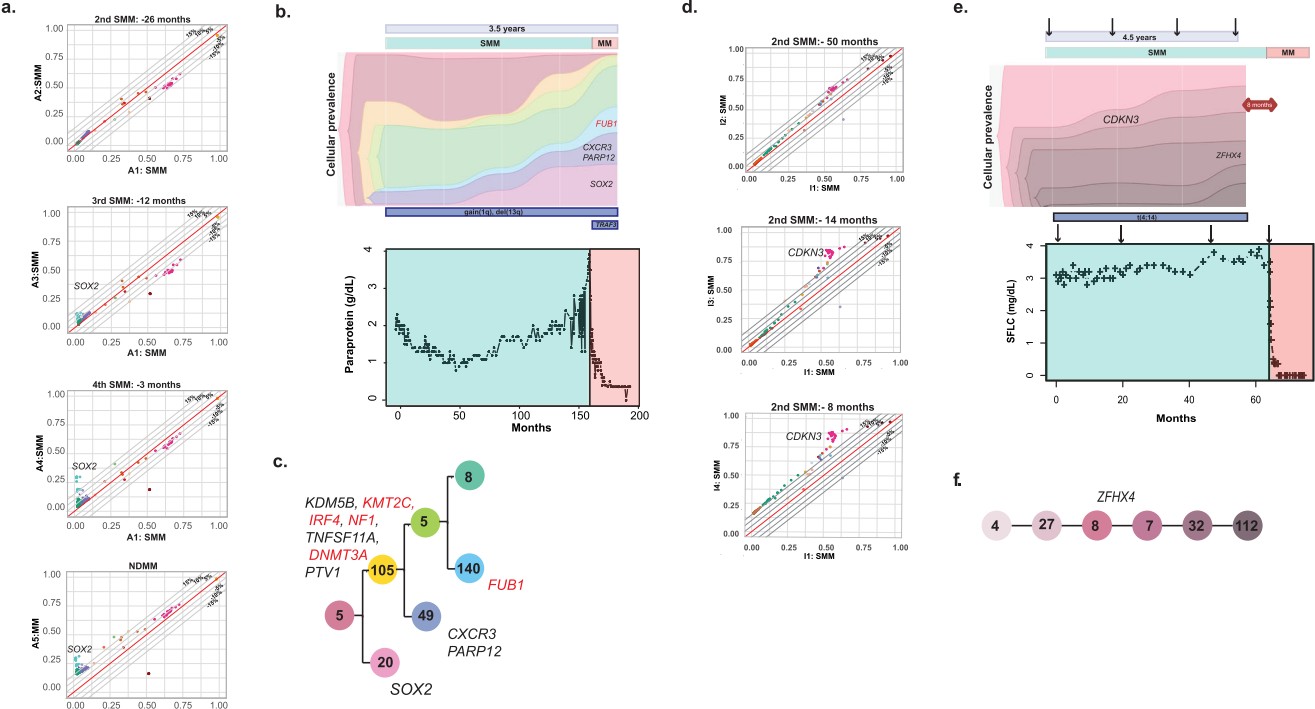

**Fig. 8 Genomic evolution of samples from Patient A and I. a** CCF plot of patient A showing the emergence of a *SOX2* cluster before progression. Patient and sample number indicated on axis. The dotted lines represent the % of the difference in CCF from the angle bisector line, which represents perfect identity between the samples Each color represents one cluster of mutations. **b** Fish-plot summarizing the clonal evolution in parallel to the paraprotein evolution in patient A. **c** Phylogeny tree showing branching evolution of patient A. Colors correspond to Fish-plot. Numbers represent the number of additional mutations in each clone CCF = cancer clonal fraction. **d** CCF plot showing the emergence of a *CDKN3* cluster before progression in patient I. Patient and sample number indicated on axis. **e** Fish-plot summarizing the clonal evolution in parallel to the paraprotein evolution in patient I. **f** Phylogeny tree showing linear evolution in patient I. Colors correspond to Fish-plot. Numbers represent the number of additional mutations in each clone CCF = cancer clonal fraction.

In order to understand the mechanistic basis driving mutational abnormalities at these early truncal stages of the disease, we used the mutation data to identify signatures and studied their relationship over time. We identified the background signature SBS1 and 5 and show that these are stable over time and were not influenced by their relationship to the time of transformation.

In 2015, we first reported a significant contribution of APO-BEC hypermutation signatures in the *MAF* and *MAFB* translocated newly diagnosed subgroups, where they made-up a median of 58% and 44% of the total mutation in the t(14;16) and t(14;20), respectively. We show that the proportion of APOBEC mutations is higher in MM than in SMM with a trend toward lower levels in SMM mafs than MM mafs. These observations are consistent with there being two distinct levels of APOBEC signatures in MM. These data suggest that an APOBEC mutational signature may be associated with progression from SMM to MM, in keeping with this, we observe a trend towards more rapid progression to MM in cases with an APOBEC signature over the level of 5%.

The evolutionary relationship of sub-clones in precursor stages is critical to disease development and its clinical behavior. While prior reports have studied paired pre- and post-progression samples none have had access to multiple sequential samples. In contrast, in cases of SMM increasing diversity did occur over time reflecting the ongoing acquisition of a more complex clonal structure with time in this group. Overall, these observations are consistent with the idea that applying the Shannon diversity index to serial samples may be able to identify cases at higher risk of progression but will need confirmation in a larger data set

(Fig. 8). We show that changes in clonal substructure can be used to monitor SMM before end-organ damage develops[17]. Our findings are consistent with those presented by Bustoros et al where they also note pre-existing sub-clonal heterogeneity. They recognize changes in sub-clonal CNA that were associated with clonal expansion between timepoints. However, the caveat is that they had only two timepoints and thus were unable to identify clonal tiding. The current analysis has a greater ability to detect changes in sub-clonal structure and indicates that branching evolution is the predominant pattern of progression.

The results of our sequential analysis clearly show that an increase in a sub-clonal fraction occurs before biochemical progression and the development of end-organ damage. This observation is a critical advance because with such changes preceding clinically relevant events by more than 1 or 2 years a safe therapeutic window can be defined. Such an approach could prevent the potential adverse impact of therapeutic intervention and would restrict intervention to a time point associated with the maximal risk-benefit ratio for the patient without significantly impair quality of life in an ongoing fashion.

Knowledge of the genetic basis of SMM can allow us to better define and monitor early disease stages. The results of this study suggest that the presence of only a limited number of drivers typical of MM can identify SMM that are destined to rapidly develop MM (Fig. 9). Going forward with improved definitions of SMM it may be possible to identify a group of SMM where treatment is indicated based on the presence of *KRAS* mutations, APOBEC signatures, and monitoring sub-clonal structure over time.

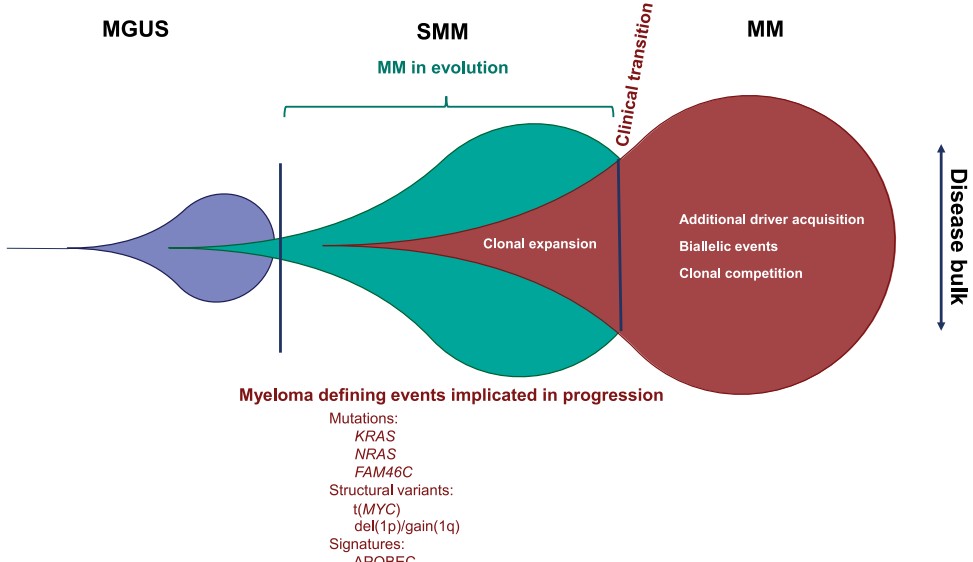

**Fig. 9 The distinct molecular effectors at play in the progression from SMM to MM.** SMM progresses to MM after acquiring a series of secondary events such as key mutations, structural events, biallelic events, or APOBEC signatures that drive progression.

## Methods

**Patients and samples**. Eighty-two previously untreated SMM patients according to IMWG 2014 criteria[8] were included in the study from the University of Arkansas for Medical Science after written informed consent. The median follow-up was 5.18 years (95% CI 3.53–6.59) from diagnosis. An additional 223 previously published[30] newly diagnosed MM patients were used for comparison. Clinical data were collected and checked for consistency. A summary of patients' characteristics in each cohort may be found in Table 1. Ten early MM (EM) patients with new early myeloma criteria only (bone marrow plasma cell ≥60%, SFLC ratio ≥100, or more than one Focal lesion on MRI), and 17 MGUS were used for additional comparison.

CD138$^+$ plasma cells were isolated from bone marrow aspirates by magnetic-activated cell sorting using the AutoMACS Pro (Miltenyi Biotec GmbH, Bergisch Gladbach, Germany) or RoboSep (STEMCELL Technologies, Vancouver, Canada). Plasma cell purity was determined by flow cytometry and only samples with >85% purity were used in this study. DNA from peripheral blood or stem cell harvest was used as a matched non-tumor control sample for each patient to exclude germline variants. Nucleic acids were isolated using the AllPrep DNA/RNA or Puregene kits (Qiagen, Hilden, Germany).

### Data processing, variants calling, filtering, and annotation

*Targeted panel*. Samples were processed as previously published[30]. In brief, we performed targeted sequencing on 125 genes and chromosomal regions that had previously been shown to be relevant to the biology, prognosis, and treatment of MM. This included the tiling of the Ig regions and *MYC* to identify Ig translocations, the V, D, and J rearrangements[27,31,32], and *MYC* abnormalities.

The panel was divided into a translocation panel and a mutation/copy-number panel to provide high depth coverage for mutation analysis (0.6 Mb) while providing lower depth sequencing of translocation regions (4.2 Mb). Each patient had their tumor and control DNA sequenced, to identify somatic mutations, CN changes, and translocations. The median mean coverage of each panel may be seen in Supplemental Table 3. Sequencing data were analyzed as previously described[30]. The pipeline is available on Github[32].

*Whole-exome sequencing*. 62 samples (9 controls and 53 samples) underwent custom-enriched exome sequencing. 200 ng of DNA was used to prepare libraries using the HyperPlus kit (Kapa Biosystems) and hybridized to a modified SeqCap EZ MedExome (Roche Nimblegen), which included Ig and *MYC* regions for translocation detection. Cross sample contamination was assessed by SNP mismatch analysis. The median coverage was 93× (IQR 68-128) and 100× (IQR 95-103) for tumors and controls, respectively.

The files were demultiplexed using bcl2fastq and aligned to the Ensembl hg38 reference genome using BWA mem (v. 0.7.12) (variants were called using Strelka (v.1.0.14) and single nucleotide variants (SNVs)) were filtered using fpfilter (https://github.com/ckandoth/variant-filter). A 10% cut-off was used to filter Indels. Annotation was performed using Variant Effect Predictor (v.85). Sequenza v3.0.0[33] was used to detect somatic copy-number aberration and estimate tumor purity and ploidy. Finally, intra- and inter-chromosomal rearrangements were called using Manta (v0.29.6) using the default settings and the exome flag specified.

We attempted to reconstruct the clonal population structure of WES samples that had adequate CNA ($n = 44$) using Pyclone[34]. PyClone is a Bayesian clustering method for grouping sets of deeply sequenced somatic mutations into putative clonal clusters while estimating their cellular prevalences and accounting for allelic imbalances introduced by segmental copy-number changes and normal-cell contamination. All samples with appropriate CNA were used. Data were visualized using the Fish-plot package[35].

*Ultra low-pass whole-genome sequencing*. Sixty-nine SMM and 116 MM samples underwent ultra low-pass WGS. Libraries for tumor DNA and control DNA were prepared as described above using the HyperPlus kit (Kapa Biosystems). Before hybridization to the panels, library DNA was removed and sequenced directly using paired tumor and control libraries, which were pooled for sequencing on the NextSeq500 using 75-bp single-end reads. The average coverage was 0.29 (IQR 0.16–0.51) and 0.21 (IQR 0.18–0.3) for SMM and MM, respectively.

Sequence reads were aligned to Ensembl GRCh37/hg19 and the copy number was determined using Control-FREEC (v3.0.0).

### Gene expression scores

Total RNA from plasma cells was used for gene expression profiling (GEP) using U133 Plus 2.0 microarrays (Affymetrix). CEL files were normalized using GCRMA[36] for the application of the updated TC algorithm. MAS5 normalization was also performed when necessary, e.g., for calculation of GEP4 and NF-κB scores. All expression data were normalized using R Bioconductor and transformed to the UAMS TT2 and TT3 NDMM standard according to a variant of M-ComBat[37].

### Droplet Digital PCR (ddPCR)

*Primers*. Primers were used to amplify the *IGHG3-MYC* translocation and the non-translocated *IGH* locus. Primers and probes may be found in Supplemental Table 5.

*ddPCR reactions*. 25 μL reaction mixtures were prepared containing primers (40×, ThermoFisher Scientific), 10 ng template, and ddPCR™ Supermix (2×, Bio-Rad). Droplet generation and transfer of emulsified samples to PCR plates were performed according to the manufacturer's instructions (Instruction Manual, QX200™ Droplet Generator—Bio-Rad). The cycling protocol started with a 95 ℃ enzyme activation step for 10 min followed by 40 cycles of a two-step cycling protocol (94 ℃ for 30 s and 60 ℃ for 1 min). The ramp rate between these steps was slowed to 2 ℃/s. The sample was read using a QX200 droplet reader (Bio-Rad). The absolute number of positive droplets was calculated using QuantaSoft (v.1.7.4).

### Quantification and statistical analysis

*Time-to-event analysis*. Time-to-event analysis was performed in R with all genetic events with $n > 8$. The Kaplan–Meier estimator was used to calculate time-to-event distributions. To determine a cut-off value for the APOBEC contribution, we performed an independence response test using maximally selected rank statistics (Maxtest)[38]. Stepwise Cox regression[39] using previously published risk factors (GEP4, and IMWG 2018 criteria[40]), and potential novel factors (del(6q), del(13q), and *KRAS* mutation) was performed. Only the final model was plotted.

*Comparison testing.* Kruskal–Wallis or Fisher's exact tests were used to compare the median of a continuous variable or the distribution of discrete variables across groups, when appropriate. Young's correction was used when appropriate. All *p*-values are two-sided, if not specified otherwise.

*Signature analysis.* Mutational signatures were called using the non-negative matrix factorization approach (NMF), which determined for each sample, the counts for the SNV types (6 possibilities) and the 3-base sequence contexts (16 possibilities). The R package "NMF" was used for all calculations. To determine the number of signatures, we ran 50 iterations of the algorithm for 2–7 signatures and chose the number that maximized both the dispersion values and cophenetic distance. The analysis was then repeated using 1000 iterations for the number of signatures previously determined, Finally, we used the cosine similarity to determine which previously published Sanger signatures best fitted the extracted signatures.

The fitting algorithm mmSig[41], which fits the mutations identified in each patient with the mutational signatures that have previously been reported in MM was used to determine the signature admixture in each sample and among samples more than 2 years away from progression, samples that were within 2 years of progression, and samples of patients that had not progressed. Based on the cosine similarity between the original mutation profiles generated without that signature, we deduced the corrected contribution of each signature to the mutational profile of each patient or group.

*Diversity analysis.* The Shannon diversity index (*H*) is an index that is commonly used to characterize species diversity in a community. Shannon's index accounts for both abundance and evenness of the species present. The proportion of species *i* relative to the total number of species ($p_i$) is calculated, and then multiplied by the natural logarithm of this proportion ($\ln(p_i)$). The resulting product is summed across species, and multiplied by −1.

$$H = -\sum_{i=1}^{R} p_i \ln(p_i).$$

**Ethics.** This study was approved by the Institutional Review Boards (IRB) of the University of Arkansas for Medical Science (#261281). All research was conducted in accordance with the Declaration of Helsinki.

**Reporting summary.** Further information on research design is available in the Nature Research Reporting Summary linked to this article.

## Data availability
The targeted panel data are deposited in the EGA database under accession code EGAD00001005056. The whole-exome sequencing are deposited in the EGA database under accession code EGAD00001005285. These data are available under restricted access, access can be obtained by contacting Gareth Morgan. The remaining data are available within the Article, Supplementary Information, or available from the authors upon request.

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

## Acknowledgements
We thank all the patients and their families for their contributions to this study. We acknowledge the continued support from the Multiple Myeloma Research Foundation and the Perelman Family Foundation. BAW and GJM received grant support through a Translational Research Program award from the Leukemia & Lymphoma Society (6020-20). E.M.B. received funding from the Fédération Françaisssssse de Recherche sur le Myélome et les Gammapathies under the aegis of the Fondation de France. We would like to acknowledge Philip Farmer and Michael Rutherford for assembling the database at UAMS.

## Author contributions
Study conception B.A.W., G.J.M., and E.M.B. Study design: B.A.W., G.J.M., and E.M.B. Acquisition of data: E.M.B., S.D., R.T., C.A., Y.W., M.A.B., C.P.W., S.T., and M.Z. Analysis E.M.B., E.H.R., F.M., F.E.D., G.J.M., and B.A.W. Interpretation of data: E.M.B., E.H.R., F.M., F.E.D., G.J.M., B.A.W., S.K.J., T.F., C.D., F.Z., F.R., C.S., A.A., B.B., and O.L. Drafting of the manuscript: E.M.B., G.J.M., and B.A.W. Critical revisions and approval: all authors.

## Competing interests
E.M.B. discloses lecture fees from Janssen, Abbvie, and Celgene; discloses travel fees from Amgen, and Celgene; none in relation to this paper. The remaining authors declare no competing interests.
