## [Peer Review File · Nature Communications]

Reviewers' Comments:

Reviewer #1:

Remarks to the Author:

The authors studied the molecular make up of smoldering myeloma by performing targeted sequencing and ULP WGS on 82 patients with SMM. In addition, they studied 9 patients with sequential BMBx. The authors report a lower frequency of NRAS, FAM46C, copy number abnormalities in SMM compared to symptomatic myeloma patients. In addition, KRAS mutations were independently associated with the risk of progression to symptomatic disease. The analysis of sequential samples from 9 patients noted a branching evolutionary pattern as well as the acquisition of novel mutations over time.

Two recent papers (one by Bustoros et al and one by Manzoni et al) have studied the genomic landscape of smoldering myeloma. the paper by Manzoni et al was limited by a smaller number of SMM patients (25) and this limits the conclusions. The paper by Bustoros et al. included 214 SMM patients and also studied sequential samples. It concluded that RAS mutations , DNA repair pathway and MYC alternation (including translocation and copy number abn) were independently predictors of progression to symptomatic myeloma.

Overall, in some ways, this manuscript confirms the importance of RAS alternation as a predictor of progression for SMM.

1- The authors note fewer t(4;14), MYC translocations, and some copy number abnormalities (del 1p, del 17p) in SMM patients. the authors postulate that this may represent a "late event" in MM progression to symptomatic disease. Another conclusion could be that such genomically high risk myeloma may evolve with a shorter smoldering / asymptomatic phase (which is underrepresented in such retrospective studies) but that the acquired genomic changes were early.

2- Aside of K/N RAS mutations, the genomic landscape of MM includes a number of mutations with low frequency (<4-5%). This makes it hard to derive conclusions with respect to each mutation when the sample size is limited to 82 patients. The authors should acknowledge this limitation as it possible that the frequency of the mutation in symptomatic MM is so low that it is hard to conclude its absence is relevant in SMM when at most you expect 3 or so patients out of 82 (line 146 for example) or it would be difficult to conclude significance of a few events (such as CCND1 and ATM observed in 2 and 1 patients who were IMWG high risk) line 186.

3- line 149 please list the observed frequency of KRAS, NRAS and FAM46C in the symptomatic MM group and SMM (for NRAS and FAM46C).

4- the authors note that mutations processes are stable at the SMM / MM interface, however this doesn't exclude that while non driver mutations may not impact the transition to overt MM, some driver mutations might and these may be low frequency events that could be missed by a more generalized analysis.

5- the author find that the patients who had a high clonal diversity index were more likely to progress to MM and this may be useful to potentially identify patients at higher risk of progression. Does this diversity index precede clinical changes observed in SMM? does it correlate with "high risk" genomic landscape?

6- the authors opine that early identification of patients likely to progress in the next 1-2 years may result in a therapeutic window where early therapy can be given. While I agree this knowledge is helpful from a biology perspective, this statement must be weighed against the main disadvantage of early therapy which undoubtedly results in a longer duration of therapy and possible resultant increased side effects, decreased quality of life and increased cost of therapy. Importantly current clinical factors such as the new IMWG definition of newly diagnosed myeloma already allow therapy for patients with a high risk of progression at 2 years. However in practice , not all such patients start therapy early for a variety of reasons.

Reviewer #2:

Remarks to the Author:

In the present manuscript Boyle et al. have explored to identify the key molecular markers underlying the progression of smoldering multiple myeloma (SMM) by ultra-low-pass whole genome sequencing. The authors showed that the KRAS mutations were associated with a shorter time to progression, and that branching evolutionary patterns, novel mutations, biallelic hits in tumour suppressor genes, and segmental copy number changes are key mechanisms underlying the transition to multiple myeloma (MM).

SMM is an asymptomatic plasma cell disorder, the behavior of which is distinguished from monoclonal gammopathy of undetermined significance (MGUS) by a higher tumour burden and rate of progression to symptomatic MM. Managing the premalignant phases of MM is becoming a major clinical goal and the key to doing this safely and effectively is to define the optimum time for intervention.

In the present study, the authors provided key mechanisms underlying the transition to MM, which can precede progression and be used to guide early intervention strategies. Although some data are confirmatory for what already reported within the Literature (Busters et al. J Clin Oncol, 2020, 38; 2380(ref.21)), prior reports have studied paired pre- and post-progression samples. In contrast, the authors used multiple sequential samples from early disease stage of SMM and the mutation data to identify signatures, and studied their relationship overtime. This leads to improve the resolution of the definitions of sub-clonal structure. Indeed, the authors showed the importance of branching evolutionary pathways at asymptomatic disease stage. The reviewer thinks that this observation is a critical advance and would be of interest to others in community.

Minor points

1. In contrast to prior reports (Bolli et al, Nat Commun, 2018,9; 3363(ref. 15), Busters et al. J Clin Oncol, 2020, 38; 2380(ref.21)), the authors were unable to identify the APOBEC signature at significant level. Discuss with the difference.
2. Figure 4. B; the authors should show what the arrows indicate. C; the authors should show which line indicates MYC/IGHG3 ratio, Bone marrow plasma cells, or M-spike. In the manuscript, the authors described the legend of Figure 4D (line 752, p27), however, the reviewer could not find it in Figure 4.
3. Reference 27 (line 662, p23). J Clin Oncol. 2015; 33(33): 3911-3920.

Reviewer #3:

Remarks to the Author:

Boyle et al 'The Molecular Make Up of Smoldering Myeloma Highlights the Evolutionary Pathways Leading to Multiple Myeloma.'

Summary

- This paper describes key genomic findings from an investigation into the molecular drivers and sub-clonal architecture of smoldering myeloma (SMM) and its progression to multiple myeloma (MM). It sheds light on the genomic basis and sub-clonal architecture of SMM compared to MM, and the identification of cases at risk of progression to MM based on this genomic data. The major points of the paper are:
 - ♣ The presentation of key genomic similarities and differences between a cohort of SMM and MM, specifically, that mutations in NRAS and FAM46C and adverse translocations are less common in SMM than MM.
 - ♣ KRAS mutations are associated with shorter time to progression of SMM to MM
 - ♣ Detailed presentation of genomic changes in multiple sequential samples of SMM in 9 patients identifying mechanisms underlying progression to MM
 - ♣ The results may lead to improved monitoring of SMM and inform the early detection of patients with SMM who will progress to MM

- The authors performed NGS on a cross-sectional study of 82 untreated SMM cases as diagnosed by IMWG 2014 criteria. These findings were compared to a previously published dataset of 223 MM cases, studied using the same targeted panel. Ultra-low pass whole genome sequencing used to obtain copy number information on a subset of 68 SMM and 116 MM patients was also undertaken. In addition, a sequential study of 9 patients with a total of 53 samples was performed.
- The authors first present a description of the type and frequency of translocations, copy number abnormalities, mutations and biallelic loss identified in SMM and compared to MM. Factors associated with time to progression of SMM to MM were next assessed and KRAS mutations, and high-risk GEP4 or IMWG score were found to be associated with a shorter time to progression on multivariate analysis. A comparison of gene signatures between SMM, MM, MGUS and normal controls to assess activity of the NFκB is next described. The subsequent sections focus on the cohort of sequential samples and describe the genetic changes in these samples over time, including at progression to MM. Structural abnormalities, mutation load, mutational signatures, sub-clonal architecture and clonal diversity are all described.
- Overall this data dense manuscript could be improved with revisions and synthesis in the discussion.

Comments

General:

1. A large sequencing study on smouldering myeloma was published recently: Bustoros et al 'Genomic Profiling of Smoldering Multiple Myeloma Identifies Patients at a High Risk of Disease Progression' J Clin Oncol, July 2020. This study has a larger cohort size, uses similar techniques and has some similar conclusions to the study under review. However, the unique aspect of the paper under review is the more detailed work on the sub-clonal architecture of SMM through serial sampling. The previous study had five cases with two sequential samples each, collected at least 12 months apart, while the study under review had nine patients, with a total of 53 individual samples and further analytical tools applied. The current authors reference the earlier paper in their results and in their discussion, but the differences and similarities could be accentuated further in the discussion.

Introduction:

1. Page 3, lines 61 – 71: the way this is written suggests that the IMWG took the comments from reference 4 into consideration when determining their diagnostic criteria for SMM, which is temporally implausible. Additionally, the IMWG 2014 don't refer to 'early myeloma' and there is no formal definition of this entity. In contrast, there is a formal definition of SMM provided by the IMWG 2014 and reinforced by the WHO 2017 Classification of Tumours of Haematopoietic and Lymphoid Tissues. This section should be revised for clarity, and a clearer working definition for the term 'EM' provided.

Methods:

1. Supplemental Table 4 is not present in the data supplement
2. The rationale for whole exome sequencing in addition to the targeted panel and whole genome sequencing is not clear. Presumably it was for mutation load determination, or clonal architecture study, however this is not explained in the Methods or main text and should be addressed to aid understanding as to why it was performed.
3. There are limited details provided for the gene expression methodology and scoring. Two references are provided (36, 37) however if word limit allows, including additional basic information (ie: Affymetrix array) would ease the understanding of this section. Additionally, the abbreviation 'TC' is used in this section without explanation.
4. There are limited details regarding sample inclusion, in particular, where it says 'samples kept if they had a depth of >50' line – what percentage of the panel had to be covered at >50x for them to be kept, versus rejected? It is important to clarify this, as it contributes to sensitivity for the detection of variants.
5. It would aid understanding of the manuscript re ultra-low pass WGS metrics to state the amount of the genome covered to a particular depth (e.g 5x) and to know what proportion of the genome was assessed for variants and at what sensitivity.

Results:

1. Page 5, line 139: 'we next investigated the impact and frequency' – this paragraph provided a description on the frequency of driver mutations in SMM vs MM however no discussion on the impact of these mutations was provided, rephrase please.

2. Page 7 and 8, lines 189, 205 and 216: the n=7 and n=10 cut-offs for univariate and multivariate analysis seem arbitrary; what was the rationale for choosing these cut-offs?
3. Page 8, line 226: 'lesion' – does this refer to 'bony lesion'? Please update for clarity.
4. The abbreviation 'NDMM' is not introduced within the text, only in the Figure 1 caption. Similarly, the abbreviations 'HRD' and 'CCF' are not introduced at the first use. This should be rectified for ease of reading.
5. Page 12, from line 343 regarding Shannon's diversity index – the sample numbers (n=2 vs n=6) and effect sizes (median ~1.4 for not progressed vs ~1.66 for progressed) are quite small. I think a caveat regarding the small sample size etc is warranted in the main text to avoid over interpretation/emphasis of these findings. Additionally, the conclusion on lines 350 – 353 are unclear. As written, it suggests that cases with a high and stable diversity index have already transformed...Suggest review of this section for clarity and de-emphasis of the findings due to small sample number and effect size.

Discussion:

Line 425 – "We show that these changes in clonal substructure can be used to monitor the asymptomatic stage of SMM end organ damage has developed." rephrase
 APOBEC driven processes – were briefly discussed but why is it discordant with the JCO data?
 I think the authors could synthesize things better, using the concepts they have pooled in fig 8, creating the vision/ outline if these findings can be reproduced for the future management of these pts.

Figures and Tables:

1. Supplemental Table 1 contains two entries for del(16q) with different values. Suggest revise.
2. Supplemental Table 2 exists twice (page 3 and 4), as separate tables: 'Summary of sample sequencing depth' and 'List of genes on the targeted panel.'
3. Supplemental Figure 1 appears to have an errant set of values, top right: ' $\chi^2 = 43$ p<0.0001'
4. Supplemental Figure 2 – the number of TRAF3 mutations in "EM" are significantly more than MM - why -please discuss
5. Supp Fig 5E – mut/del17p mentioned in text but not in the figure
6. Supplemental Figure 8 – concept is fine, but not clearly depicted, and really hard to interpret
7. Figure 2B – seems to be the same data as supplemental figure 4.
8. Figure 4 C is difficult to interpret: the Y axis has two labels; the figure legend appears incomplete (the key to the coloured lines is absent), and the label above the plot 'detected by NGS' is of unclear relevance to the figure and data.
9. Figure 4a - difficult to read the progressions – could be made clearer
10. Figure 8 'Signatures' box contains the word 'clock.' This is not discussed or defined anywhere in the text, ?in error. CNV could be added to this figure perhaps??

Reviewer #1 (Remarks to the Author):

1- The authors note fewer t(4;14), MYC translocations, and some copy number abnormalities (del 1p, del 17p) in SMM patients. the authors postulate that this may represent a "late event" in MM progression to symptomatic disease. Another conclusion could be that such genomically high risk myeloma may evolve with a shorter smoldering / asymptomatic phase (which is underrepresented in such retrospective studies) but that the acquired genomic changes were early.

We agree and have modified the text accordingly. Line 369: "An Alternative interpretation may be that high-risk myeloma evolves more rapidly through a shorter smoldering phase thus explaining why it is under-represented in SMM."

2- Aside of K/N RAS mutations, the genomic landscape of MM includes a number of mutations with low frequency (<4-5%). This makes it hard to derive conclusions with respect to each mutation when the sample size is limited to 82 patients. The authors should acknowledge this limitation as it possible that the frequency of the mutation in symptomatic MM is so low that it is hard to conclude its absence is relevant in SMM when at most you expect 3 or so patients out of 82 (line 146 for example) or it would be difficult to conclude significance of a few events (such as CCND1 and ATM observed in 2 and 1 patients who were IMWG high risk) line 1

We agree and have added this information to the text: Line 139: ". A comparison of rare mutations was not possible given the sample size and so we restricted our analysis to the comparison of mutations occurring in known myeloma relevant pathways."

3- line 149 please list the observed frequency of KRAS, NRAS and FAM46C in the symptomatic MM group and SMM (for NRAS and FAM46C).

The relevant frequencies have been added:

- KRAS: 22% in MM versus 13% in SMM
- NRAS: 17% in MM versus 4.5% in SMM
- FAM46C: 7% in MM versus 0% in SMM

Line 135: The most frequently mutated gene in SMM was *KRAS* (n=11, 13.4%). In comparison to NDMM, fewer *NRAS* (4.5% versus 17%, $\chi^2=6,4$, $p=0.01$), and *FAM46C* (0% versus 7%, Fisher's, $p=0.008$) mutations were detected and there was a trend towards fewer *KRAS* (13% vs 22%) mutations in SMM, **Figure 2A and Supplemental Figure 2.**

4- The authors note that mutations processes are stable at the SMM / MM interface, however this doesn't exclude that while non driver mutations may not impact the transition to overt MM, some driver mutations might and these may be low frequency events that could be missed by a more generalized analysis.

We agree with the reviewer that we do not see mutational signatures changing at the transition from SMM to MM in the serial samples studied, however there could be individual driver mutations in key genes that are associated with progression at low frequency. Despite this, the algorithm used to define mutational signatures is not sensitive to individual mutational events. We address this earlier in the manuscript by showing that *NRAS* and *FAM46C* were under-represented in the SMM group and may be mutational drivers of progression that are not frequent enough to be considered a signature.

5- the author find that the patients who had a high clonal diversity index were more likely to progress to MM and this may be useful to potentially identify patients at higher risk of progression. Does this diversity index precede clinical changes observed in SMM? does it correlate with "high risk" genomic landscape?

This is a very interesting question that we can only partially address given the sample size. Overall, the Shannon diversity was not significantly different between samples that carried *KRAS* mutations, or high-risk features such as gain(1q) or t(4;14). This information has been added to supplemental figure 16 and the text modified accordingly.

Line 347: There was no difference between patients with *KRAS* mutations or high-risk features (t(4;14) and gain(1q)).

the authors opine that early identification of patients likely to progress in the next 1-2 years may result in a therapeutic window where early therapy can be given. While I agree this knowledge is helpful from a biology perspective, this statement must be weighed against the main disadvantage of early therapy which undoubtedly results in a longer duration of therapy and possible resultant increased side effects, decreased quality of life and increased cost of therapy. Importantly current clinical factors such as the new IMWG definition of newly diagnosed myeloma already allow therapy for patients with a high risk of progression at 2 years. However in practice, not all such patients start therapy early for a variety of reasons.

We agree with the reviewer. This point has been highlighted in recent discussion within the IMWG that concluded that prolonged therapeutic interventions were not a therapeutic requirement in these populations. In this work, we have highlighted a population that is still at high risk of progression despite the novel IMWG criteria which could add precision to the risk assessment. We believe it is up to the clinician to weigh the benefit and the risk of early intervention based on a thorough personalized risk-assessment strategy, all this paper is doing is offering additional tools to best do this. We have rephrased the discussion to highlight this: line 361

Line 360: ~~Further, strengthening the definitions of SMM cases suitable for intervention could improve the health care value of early intervention.~~ Further, adding genetic factors into personalized risk-assessment tools may improve treatment initiation decisions..

Reviewer #2 (Remarks to the Author):

Minor points

1. In contrast to prior reports (Bolli et al, Nat Commun, 2018,9; 3363(ref. 15), Busters et al. J Clin Oncol, 2020, 38; 2380(ref.21)), the authors were unable to identify the APOBEC signature at significant level. Discuss with the difference.

We thank the reviewer for this comment. Regarding the APOBEC signature, we first identified this mutational signature enriched in the t(14;16) and t(14;20) myeloma subgroups (collectively termed the maf group) in our Nature Communications paper in 2015 (doi: 10.1038/ncomms7997), where it comprises a median of ~60% of mutations in the t(14;16) samples but only ~10% in non-maf samples. It is, therefore, making up the majority of mutations in the maf subgroup. In SMM we saw a significantly smaller APOBEC contribution than in MM. Furthermore, in maf SMM patients this contribution is 17%, compared to 11% in non-maf samples. Therefore, we do not identify any significant difference between the two subgroups. The contribution in maf SMM samples also appears lower than that of maf MM samples, 17% vs. 41%, suggesting that it is not present at significant levels in the asymptomatic state. Bustoros et al. do not specify the percentage contribution of the APOBEC signature in their subgroups and only mention APOBEC-associated mutations in 14 genes rather than the contribution of the signature across all mutations, which probably indicates that they did not see any difference across the entire dataset. Therefore, our data are in agreement in that we both believe the APOBEC mutational process is linked to progression from SMM to MM.

We have added to the result section and the discussion:

Line 148: In terms of mutational signatures, we show that the contribution of APOBEC is significantly lower in SMM than in MM (11% (0-43%) versus 17% (0-48%), $\chi^2=5.2$, $p = 0.02$) consistent with it having a role later in disease progression, **Figure 3A**. Furthermore, in SMM patients with either a *MAF* or *MAFB* translocation (termed maf), the median APOBEC contribution is 18% (0-54%, $n=4$), compared to 11% (0-43%, $\chi^2=0.5$, $p=0.4$, $n=78$) in non-maf samples. Therefore, unlike observations in MM, we do not identify any significant difference between the two subgroups in SMM. Finally, the APOBEC contribution also seems lower in maf-SMM than it does in maf-MM (16% (0-44%, $n=4$) vs. 41% (0-100%, $n=15$)). Despite the small sample size, these data suggest that APOBEC is associated with disease progression, **Figure 3B- Supplemental Figure 3A**.

Line 200: Finally there was a trend suggesting that patients with a small contribution of APOBEC signatures (<5%) had a better outcome than the others (**Supplemental Figure 3B**).

Line 314: ~~We specifically looked for APOBEC driven processes as these have been previously reported in SMM,⁴ but, using the criteria we used previously to define this signature in cases of t(14;16), we were unable to identify the APOBEC signature at significant level.~~ In 2015, we first reported a significant contribution of APOBEC hypermutation signatures in the MAF and MAFB translocated newly diagnosed subgroups, where they made-up a median of 58% and 44% of the total mutations in the t(14;16) and t(14;20) respectively. We show that the proportion of APOBEC mutations is higher in MM than in SMM with a trend toward lower levels in SMM mafs than MM mafs. These observations are consistent with there being two distinct levels of APOBEC signatures in MM. These data suggest that an APOBEC mutational signature may be associated with progression from SMM to MM, in keeping with this we observe a trend towards more rapid progression to MM in cases with an APOBEC signature over the level of 5%.

2. Figure 4. B; the authors should show what the arrows indicate. C; the authors should show which line indicates MYC/IGHG3 ratio, Bone marrow plasma cells, or M-spike. In the manuscript, the authors described the legend of Figure 4D (line 752, p27), however, the reviewer could not find it in Figure 4.

We agree with the reviewer's comment and have adjusted the figure accordingly. The arrows highlight changes in copy number. This has been added to the legend. The legend of each color line in 4C has been added. Figure 4D was removed prior to submission. The legend now reads:

Figure 6: Acquisition of drivers in SMM patients over time. A. Swimmer plot of the group of patients (A-G) followed over time. The hatched portion of each bar represents the SMM phase of disease and the unhatched portion represents the MM phase of

disease. B. Plot showing changes in copy number over time with a focus on loss of del(5) and the acquisition of gain(1q), del(11q) and del(13). Arrows highlight changes in CNA. C. The acquisition of a t(8;14) within a myeloma propagating cell leads to outgrowth of the clone until it dominates the tumor population. EM= early myeloma, FU= follow-up, NDMM= newly diagnosed myeloma, NGS= next generation sequencing, MS= M-spike.

3. Reference 27 (line 662, p23). J Clin Oncol. 2015; 33(33): 3911

This has been corrected.

Reviewer #3 (Remarks to the Author):

Comments

General:

1. A large sequencing study on smouldering myeloma was published recently: Bustoros et al 'Genomic Profiling of Smoldering Multiple Myeloma Identifies Patients at a High Risk of Disease Progression' J Clin Oncol, July 2020. This study has a larger cohort size, uses similar techniques and has some similar conclusions to the study under review. However, the unique aspect of the paper under review is the more detailed work on the sub-clonal architecture of SMM through serial sampling. The previous study had five cases with two sequential samples each, collected at least 12 months apart, while the study under review had nine patients, with a total of 53 individual samples and further analytical tools applied. The current authors reference the earlier paper in their results and in their discussion, but the differences and similarities could be accentuated further in the discussion.

We agree with the reviewer and have added the following to the discussion: Line 433: "Our findings are consistent with those presented by Bustoros et al where they also note pre-existing sub clonal heterogeneity. They recognize changes in subclonal CNA that were associated with clonal expansion between timepoints. However, the caveat is that they had only two timepoints and thus were unable to identify clonal tiding. The current

analysis that has a greater ability to detect changes in subclonal structure and indicates that branching evolution is the predominant pattern of progression.”

Introduction:

1. Page 3, lines 61 – 71: the way this is written suggests that the IMWG took the comments from reference 4 into consideration when determining their diagnostic criteria for SMM, which is temporally implausible. Additionally, the IMWG 2014 don't refer to 'early myeloma' and there is no formal definition of this entity. In contrast, there is a formal definition of SMM provided by the IMWG 2014 and reinforced by the WHO 2017 Classification of Tumours of Haematopoietic and Lymphoid Tissues. This section should be revised for clarity, and a clearer working definition for the term 'EM' provided.

We agree with the reviewer. The inclusion of the new IMWG criteria for MM have modified the definition of SMM by extension. The term Early myeloma was coined to highlight and allow comparability with historical datasets that used the IMWG 2009 diagnostic criteria. By not including these patients in either group we highlight their genetic resemblance to MM more than SMM and enhance the comparability of groups. The reference has been corrected and the term Early myeloma defined more precisely. To help this definition, this has been added to the introduction:

Line 63: A group with an 80% 2-year Progression-Free Survival (PFS) was identified that was redefined as MM.⁸ Here, to enhance historical comparison, we use the term Early myeloma (EM), to identify patients that fail to meet the current criteria but would have been defined as SMM previously.

Methods:

1. Supplemental Table 4 is not present in the data supplement

There was a duplicate Supplemental table 2. We regret this error and have renumbered the tables accordingly.

2. The rationale for whole exome sequencing in addition to the targeted panel and whole genome sequencing is not clear. Presumably it was for mutation load determination, or clonal architecture study, however this is not explained in the Methods or main text and should be addressed to aid understanding as to why it was performed.

We have added this information to the text to clarify:

Line 87: We performed targeted sequencing to interrogate known drivers and Ig translocations.

Line 119: we performed ultra-low pass whole genome sequencing (ULP-WGS) on a subset of patients (69 patients with SMM and 116 Newly-diagnosed MM (NDMM) patients) to determine CNA across the genome.

Line 253: we performed ultra-low pass whole genome sequencing (ULP-WGS) on a subset of patients (69 patients with SMM and 116 Newly-diagnosed MM (NDMM) patients) to determine CNA across the genome.

3. There are limited details provided for the gene expression methodology and scoring. Two references are provided (36, 37) however if word limit allows, including additional basic information (ie: Affymetrix array) would ease the understanding of this section. Additionally, the abbreviation 'TC' is used in this section without explanation.

We thank the reviewer for pointing this out. We used total RNA from plasma cells for gene expression profiling (GEP) using U133 Plus 2.0 microarrays (Affymetrix). Raw signals were MAS5 normalized using the Affymetrix Microarray GCOS1.1 software. TC refers to the translocation cyclin classification of MM and has been added to the methods.

Line 545: Total RNA from plasma cells was used for gene expression profiling (GEP) using U133 Plus 2.0 microarrays (Affymetrix). CEL files were normalized using GCRMA (Wu J & Gentry J, 2020) for application of the updated TC algorithm. MAS5 normalization was also performed when necessary, e.g. for calculation of GEP4 and NF-κB scores. All expression data was normalized using R Bioconductor and

transformed to the UAMS TT2 and TT3 NDMM standard according to a variant of M-ComBat (Stein *et al*, 2014)

4. There are limited details regarding sample inclusion, in particular, where it says 'samples kept if they had a depth of >50' line – what percentage of the panel had to be covered at >50x for them to be kept, versus rejected? It is important to clarify this, as it contributes to sensitivity for the detection of variants.

We apologize for the confusion. In this dataset we only included samples with a median depth greater than 50X for the exact reason the reviewer suggested - to maintain the sensitivity of our analysis.

We have simplified the method section accordingly which now reads:

Line 516: 62 samples (9 controls and 53 samples) underwent custom-enriched exome sequencing. 200 ng of DNA was used to prepare libraries using the HyperPlus kit (Kapa Biosystems) and hybridized to a modified SeqCap EZ MedExome (Roche Nimblegen) which included Ig and MYC regions for translocation detection. Cross sample contamination was assessed by SNP mismatch analysis. The median coverage was 93x (IQR 68-128) and 100x (IQR 95-103) for tumors and controls, respectively.

5. It would aid understanding of the manuscript re ultra-low pass WGS metrics to state the amount of the genome covered to a particular depth (e.g 5x) and to know what proportion of the genome was assessed for variants and at what sensitivity.

We apologize for the confusion. ULP-WGS was only used for copy number detection and was not used for single nucleotide variant detection. The depth of ULP-WGS was 0.7x which is not sufficient for SNV detection, but can be used for copy number detection. Using iChor it has been shown to be possible to detect copy number changes with 0.1x genome coverage in samples with down to 10% tumor content (Adalsteinsson *et al*, 2017). As we had enriched samples in this study (>70% tumor content) and a greater depth (0.7x) we are able to detect sub-chromosomal level abnormalities.

Results:

1. Page 5, line 139: 'we next investigated the impact and frequency' – this paragraph provided a description on the frequency of driver mutations in SMM vs MM however no discussion on the impact of these mutations was provided, rephrase please.

We thank the reviewer for spotting this and we have rephrased this sentence that now reads:

Line 127: We next investigated the frequency of important mutations in SMM and MM using the same targeted sequencing panel.

2. Page 7 and 8, lines 189, 205 and 216: the $n=7$ and $n=10$ cut-offs for univariate and multivariate analysis seem arbitrary; what was the rationale for choosing these cut-offs?

Thank you for this comment. We have tried to be as rational as possible when defining these complex cutoffs. Regarding the multivariable logistic regression, or any likelihood-based procedure, a key concern is potential sparse-data bias (i.e. too many covariates chasing too few outcomes) (Greenland *et al*, 2000). A general rule is that 10 events of the outcome of interest are required for each variable in the model including the exposure of interest (i.e. 10:1 events per variable). This rule may be insufficient in the presence of categorical covariates because of the increased degrees of freedom compared with dichotomous covariates. After tabulating the outcome/strata a 3 variables with $n>10$ was the optimum solution to prevent excessive overfitting.

Using $n>7$ for univariate, was the minimal number that could be used with a reasonable distribution. It was the same cutoff used by Bustoros *et. al*

We acknowledge the limitations of our sample size and believe we have discussed this in the Discussion section and have taken a cautious approach when adding variables with $n>7$ to the multivariate analysis (Supplemental data).

3. Page 8, line 226: 'lesion' – does this refer to 'bony lesion'? Please update for clarity.

We apologize for the confusion. Lesion refers to genetic lesions. To prevent confusion we have modified the sentence and replaced the word “lesion” with “molecular lesion”

Line 226: When more than one molecular lesion was present.

4. The abbreviation ‘NDMM’ is not introduced within the text, only in the Figure 1 caption. Similarly, the abbreviations ‘HRD’ and ‘CCF’ are not introduced at the first use. This should be rectified for ease of reading.

These have all been corrected.

5. Page 12, from line 343 regarding Shannon’s diversity index – the sample numbers (n=2 vs n=6) and effect sizes (median ~1.4 for not progressed vs ~1.66 for progressed) are quite small. I think a caveat regarding the small sample size etc is warranted in the main text to avoid over interpretation/emphasis of these findings. Additionally, the conclusion on lines 350 – 353 are unclear. As written, it suggests that cases with a high and stable diversity index have already transformed...Suggest review of this section for clarity and de-emphasis of the findings due to small sample number and effect size.

We agree with the reviewer and have added the following to the text.

Line 352: In contrast, in cases of SMM an increase in diversity was seen over time. These observations suggest that applying the H- indices to serial samples may be able to identify cases at higher risk of progression but need confirmation in larger datasets.

Discussion:

Line 425 – “We show that these changes in clonal substructure can be used to monitor the asymptomatic stage of SMM end organ damage has developed.” Rephrase’

We have rephrased this statement as follows:

Line 438: We show that changes in clonal substructure can be used to monitor the asymptomatic stage of SMM before end organ damage develops ¹⁷.

APOBEC driven processes – were briefly discussed but why is it discordant with the JCO data?

We thank the reviewer for this comment and have added to the result section and the discussion. Regarding the APOBEC signature, we first identified this mutational signature enriched in the t(14;16) and t(14;20) myeloma subgroups (collectively termed the maf group) in our Nature Communications paper in 2015 (doi: 10.1038/ncomms7997), where it comprises a median of ~60% of mutations in the t(14;16) samples but only ~10% in non-MAF samples (Figure 4D). It is, therefore, making up the majority of mutations in the maf subgroup. In maf SMM patients the contribution is 17%, compared to 11% in non-MAF samples. Therefore, we do not identify any significant difference between the two subgroups. The contribution in maf SMM samples is also lower than that of MAF MM samples, 17% vs. 41%, indicating that it is not present at significant levels in the asymptomatic state. Bustoros et al. do not specify the percentage contribution of the APOBEC signature in their subgroups and only mention APOBEC-associated mutations in 14 genes rather than the contribution of the signature across all mutations, which probably indicates that they did not see any difference across the entire dataset. Therefore, our data are in agreement in that we both believe the APOBEC mutational process is linked to progression from SMM to MM, it just differs in the definition of the level of APOBEC signature in the samples.

Line 148: In terms of mutational signatures, we show that the contribution of APOBEC is significantly lower in SMM than in MM (11% (0-43%) versus 17% (0-48%), $\chi^2=5.2$, $p = 0.02$) consistent with it having a role later in disease progression, **Figure 3A**. Furthermore, in SMM patients with either a *MAF* or *MAFB* translocation (termed maf), the median APOBEC contribution is 18% (0-54%, $n=4$), compared to 11% (0-43%, $\chi^2=0.5$, $p=0.4$, $n=78$) in non-maf samples. Therefore, unlike observations in MM, we do not identify any significant difference between the two subgroups in SMM. Finally, the APOBEC contribution also seems lower in maf-SMM than it does in maf-MM (16% (0-44%, $n=4$) vs. 41% (0-100%, $n=15$)). Despite the small sample size, these data suggest

that APOBEC is associated with disease progression, **Figure 3B- Supplemental Figure 3A**.

Line 200: Finally, there was a trend suggesting that patients with a small contribution of APOBEC signatures (<5%) had a better outcome than the others (**Supplemental Figure 3B**).

Line 415: ~~We specifically looked for APOBEC driven processes as these have been previously reported in SMM,⁴ but, using the criteria we used previously to define this signature in cases of t(14;16), we were unable to identify the APOBEC signature at significant level.~~In 2015, we first reported a significant contribution of APOBEC hypermutation signatures in the MAF and MAFB translocated newly diagnosed subgroups, where they made-up a median of 58% and 44% of the total mutation in the t(14;16) and t(14;20) respectively. We show that the proportion of APOBEC mutations is higher in MM than in SMM with a trend toward lower levels in SMM mafs than MM mafs. These observations are consistent with there being two distinct levels of APOBEC signatures in MM. These data suggest that an APOBEC mutational signature may be associated with progression from SMM to MM, in keeping with this we observe a trend towards more rapid progression to MM in cases with an APOBEC signature over the level of 5%.

I think the authors could synthesize things better, using the concepts they have pooled in fig 8, creating the vision/ outline if these findings can be reproduced for the future management of these pts.

We agree with the reviewers and have restructured the discussion section to illustrate each point. This now reads:

Line 454: A knowledge of the genetic basis of SMM can allow us to better define and monitor early disease stages. The results of this study suggest that the presence of only a limited number of drivers typical of MM can identify SMM that are destined to rapidly develop MM, **Figure 9**. Going forward with improved definitions of SMM it may be possible to identify a group of SMM where treatment is indicated based on the presence of *KRAS* mutations, APOBEC signatures and monitoring sub-clonal structure over time.

Figure 8. Mutational process at play appear to be none the less related to age and most of the MM specific complexity is already apparent. From a clinical perspective, monitoring sub-clonal structure over time offers a clinically useful means of directing early clinical interventions. Going forward with improved definitions of SMM it may be possible to identify a group of monoclonal gammopathy and early MM where treatment is indicated.

Figures and Tables:

1. Supplemental Table 1 contains two entries for del(16q) with different values. Suggest revise.

The gene that was used to define the slightly different region was added for clarity.

2. Supplemental Table 2 exists twice (page 3 and 4), as separate tables: 'Summary of sample sequencing depth' and 'List of genes on the targeted panel.'

We apologize and have reviewed the numbering.

3. Supplemental Figure 1 appears to have an errant set of values, top right: 'X² = 43 p<0.0001'

This is the overall Kruskal-Wallis test result This information has been added to the legend.

4. Supplemental Figure 2 – the number of TRAF3 mutations in “EM” are significantly more than MM -why -please discuss

We agree that this may lead to confusion. Given the small number of patients, these results should be interpreted with caution. To prevent this and add clarity we have revised the plot and added the number of samples per group.

5. Supp Fig 5E – mut/del17p mentioned in text but not in the figure

Only significant variables were represented in the figure and the multivariable model re-run after variable selection. This has been added to the methods at line 577.

Line 591: only the final model was plotted

6. Supplemental Figure 8 – concept is fine, but not clearly depicted, and really hard to interpret

We agree that these figures are difficult to interpret. To improve overall ease to interpret this figure we have highlight changes of interest (color bars), increased contrast, and decreased line thickness.

7. Figure 2B – seems to be the same data as supplemental figure 4.

Supplemental figure 4 and 2C are indeed very similar but differ in the ordering: 2C is ordered by GEP4 and supplemental figure 4 by IMWG risk scores. Both help highlight genes associated with high risk system. Both classifications have been added as different readers may be used to different classification system.

8. Figure 4 C is difficult to interpret: the Y axis has two labels; the figure legend appears incomplete (the key to the coloured lines is absent), and the label above the plot 'detected by NGS' is of unclear relevance to the figure and data.

We thank the reviewers for this comment. Required changes have been made to increase clarity:

- Color line added to the legend
- Color added to the y-axis label.
- Removal of the NGS line above the plot.

9. Figure 4a - difficult to read the progressions – could be made clearer

We agree with the reviewer and have added a pattern line to signify progression and altered the figure legend to read:

Figure 4: Acquisition of drivers in SMM patients over time. A. Swimmer plot of the group of patients (A-G) followed over time. The color bars represent the progression free survival. B. Plot showing changes in copy number over time with a focus on loss of del(5) and the acquisition of gain(1q), del(11q) and del(13). Arrows highlight changes in CNA. C. The acquisition of a t(8;14) within a myeloma propagating cell leads to outgrowth of the clone until it dominates the tumor population. EM= early myeloma, FU= follow-up, NDMM= newly diagnosed myeloma, NGS= next generation sequencing, MS= M-spike.

10. Figure 8 'Signatures' box contains the word 'clock.' This is not discussed or defined anywhere in the text, in error. CNV could be added to this figure perhaps??

We agree with the reviewer and have replaced "clock" by "Age-related processes" and added CAN and altered the figure to make it easier to read.

Reviewers' Comments:

Reviewer #1:

Remarks to the Author:

The authors studied the molecular make up of smoldering myeloma by performing targeted sequencing and ULP WGS on 82 patients with SMM. In addition, they studied 9 patients with sequential BMBx. The authors report a lower frequency of NRAS, FAM46C, copy number abnormalities in SMM compared to symptomatic myeloma patients. In addition, KRAS mutations were independently associated with the risk of progression to symptomatic disease. The analysis of sequential samples from 9 patients noted a branching evolutionary pattern as well as the acquisition of novel mutations over time.

Two recent papers (one by Bustoros et al and one by Manzoni et al) have studied the genomic landscape of smoldering myeloma. the paper by Manzoni et al was limited by a smaller number of SMM patients (25) and this limits the conclusions. The paper by Bustoros et al. included 214 SMM patients and also studied sequential samples. It concluded that RAS mutations , DNA repair pathway and MYC alternation (including translocation and copy number abn) were independently predictors of progression to symptomatic myeloma.

Overall, in some ways, this manuscript confirms the importance of RAS alternation as a predictor of progression for SMM.

Overall, the authors' response to the reviewers comments are appropriate.

Figure 9 seems a little bit ambiguous. Is it meant to suggests which event is earlier in the progression from SMM to MM; or is it the relative contributions of each event that is displayed? perhaps this can be clarified

Reviewer #3:

Remarks to the Author:

We believe that the Authors have tackled all the suggestions of the reviewers, and the manuscript is significantly improved.